# Rescue of *bmp15* deficiency in zebrafish by mutation of *inha* reveals mechanisms of BMP15 regulation of folliculogenesis

**Yue Zhai, Xin Zhang📍, Cheng Zhao, Ruijing Geng, Kun Wu, Mingzhe Yuan, Nana Ai, Wei Ge📍\***

Department of Biomedical Sciences and Centre of Reproduction, Development and Aging (CRDA), Faculty of Health Sciences, University of Macau, Taipa, Macau SAR, China

\* weige@um.edu.mo, gezebrafish@gmail.com

**Data Availability Statement:** All data used are provided in the paper. The RNA-seq data have been deposited in the GenBank under the BioProject No. PRJNA849009.

## Abstract

As an oocyte-specific growth factor, bone morphogenetic protein 15 (BMP15) plays a critical role in controlling folliculogenesis. However, the mechanism of BMP15 action remains elusive. Using zebrafish as the model, we created a *bmp15* mutant using CRISPR/Cas9 and demonstrated that *bmp15* deficiency caused a significant delay in follicle activation and puberty onset followed by a complete arrest of follicle development at previtellogenic (PV) stage without yolk accumulation. The mutant females eventually underwent female-to-male sex reversal to become functional males, which was accompanied by a series of changes in secondary sexual characteristics. Interestingly, the blockade of folliculogenesis and sex reversal in *bmp15* mutant could be partially rescued by the loss of inhibin (*inha-/-*). The follicles of double mutant (*bmp15-/-;inha-/-*) could progress to mid-vitellogenic (MV) stage with yolk accumulation and the fish maintained their femaleness without sex reversal. Transcriptome analysis revealed up-regulation of pathways related to TGF-β signaling and endocytosis in the double mutant follicles. Interestingly, the expression of inhibin/activin βAa subunit (*inhbaa*) increased significantly in the double mutant ovary. Further knockout of *inhbaa* in the triple mutant (*bmp15-/-;inha-/-;inhbaa-/-*) resulted in the loss of yolk granules again. The serum levels of estradiol (E2) and vitellogenin (Vtg) both decreased significantly in *bmp15* single mutant females (*bmp15-/-*), returned to normal in the double mutant (*bmp15-/-;inha-/-*), but reduced again significantly in the triple mutant (*bmp15-/-;inha-/-;inhbaa-/-*). E2 treatment could rescue the arrested follicles in *bmp15-/-*, and fadrozole (a nonsteroidal aromatase inhibitor) treatment blocked yolk accumulation in *bmp15-/-;inha-/-* fish. The loss of *inhbaa* also caused a reduction of Vtg receptor-like molecules (e.g., *lrp1ab* and *lrp2a*). In summary, the present study provided comprehensive genetic evidence that Bmp15 acts together with the activin-inhibin system in the follicle to control E2 production from the follicle, Vtg biosynthesis in the liver and its uptake by the developing oocytes.

**Funding:** This study was supported by grants from the University of Macau (MYRG2019-00123-FHS, MYRG2020-00192-FHS and CPG2023-00029-FHS) and The Macau Fund for Development of Science and Technology (FDCT173/2017/A3, FDCT0132/2019/A3 and NSFC-FDCT0086/2022/AFJ) to WG. NA and KW are supported by the FDCT Postdoctoral Funding Scheme (FDCT0029/APD/2021) and the Macau Young Scholars Program (AM2020025), respectively. The funders had no role in study design, data collection and analysis, decision to publish, or preparation of the manuscript.

**Competing interests:** The authors have declared that no competing interests exist.

## Author summary

Bone morphogenetic protein 15 (BMP15) is an oocyte-secreted growth factor that plays an essential role in female reproduction. However, scientists still do not fully understand how it works. In this study, we used zebrafish as the model to study the role of BMP15 in egg development. We genetically modified the zebrafish to delete the gene of BMP15 (*bmp15*). Without BMP15, the eggs failed to grow normally, and the female fish eventually changed sex to become males. However, if another protein, inhibin (*inha*), was also removed, the fish stayed females and their eggs could develop more normally. We found that removing inhibin changed expression of many other genes, including aromatase (*cyp19a1a*), the enzyme that makes female sex steroids (estrogens). We further demonstrated that the loss of aromatase was indeed the main reason for the failure of egg development in the fish without BMP15, and that removing inhibin helped restore the production of estrogens. In conclusion, this study found that BMP15 is vital for egg development and maintaining the female sex in fish and that it works by controlling production of female sex steroids.

## Introduction

As basic structural and functional units of the ovary, follicles in vertebrates consist of a developing oocyte and surrounding granulosa and theca cells [1]. The development of follicles, or folliculogenesis, is a multistage dynamic process involving dramatic structural and functional changes [2]. Folliculogenesis is primarily controlled by the hypothalamus-pituitary-gonad (HPG) axis [3]. Two gonadotropins from the pituitary, follicle-stimulating hormone (FSH) and luteinizing hormone (LH), play pivotal roles in regulating folliculogenesis in all vertebrates including fish [4–7]. However, it is also well known that various local factors from the ovary are also involved in regulating folliculogenesis in autocrine and/or paracrine manners [8–10]. One of the most important discoveries in female reproductive biology in the past two decades is that the oocyte serves as a controlling center during folliculogenesis by releasing a variety of peptide growth factors [11–13], among which growth differentiation factor 9 (GDF9) and bone morphogenetic protein 15 (BMP15 or GDF9B) are the best characterized. The discovery and characterization of these two factors have changed the traditional view that the oocyte is passively regulated by external factors either from the circulation or the surrounding follicle cells during folliculogenesis.

Both GDF9 and BMP15 belong to the transforming growth factor β (TGF-β) superfamily, whose members play critical roles in development and reproduction [14,15]. As the first oocyte-specific growth factor discovered, GDF9 has been well studied in both mammals and fish [16–21]. The knockout of GDF9 in mice arrested folliculogenesis at primary follicle stage, resulting in female infertility [19]. In zebrafish, the loss of Gdf9 also caused a complete blockade of follicle development at primary growth (PG) stage, similar to that in mice [22], suggesting a conserved role for GDF9 in controlling early folliculogenesis in vertebrates. Shortly after the discovery of GDF9, a second oocyte-specific member of the TGF-β family, namely BMP15, was identified in mice and humans by homology-based PCR cloning and hybridization. Both human (*BMP15*) and mouse (*Bmp15*) genes are X-linked and also expressed exclusively in the oocyte [23]. Experimental evidence has shown that BMP15 is involved in regulating the entire process of folliculogenesis from early follicle development to ovulation. In preantral stage, BMP15 maintained follicle integrity *in vitro* and promoted formation of secondary follicles [24] as well as antral formation [25]. In antral follicles, BMP15 stimulated proliferation of

granulosa cells but inhibited their luteinization as evidenced by its suppression of FSH-stimulated progesterone secretion [26]. In preovulatory follicles, BMP15 stimulated cumulus expansion and suppressed cumulus cell apoptosis [27,28], and lack of *Bmp15* gene reduced oocyte maturation [29]. At gene expression level, BMP15 suppressed FSH receptor expression as well as FSH-induced expression of LH receptor and steroidogenic enzymes [30] but induced expression of epidermal growth factor (EGF) family members and EGF receptor in the cumulus cells [27,31]. Despite these studies, knockout of the *Bmp15* gene in mice surprisingly caused sub-fertility only with reduced ovulation and fertilization rate; however, the ovarian morphology and structure were largely normal in terms of follicle development and corpus luteum formation [32]. Interestingly, the loss of BMP15 in sheep causes sterility with small ovaries containing follicles arrested at primary stage, similar to the *Gdf9* null mice [33–37]. The function of *BMP15* in fertility has also been demonstrated in humans. Mutation of *BMP15* gene has been implicated in human primary ovarian insufficiency (POI) [38,39]. The high species variation of BMP15 functions, especially between mono-ovulatory (*e.g*, sheep and humans) and poly-ovulatory (*e.g*., mice) species, raises interesting questions about its roles in other vertebrates.

In teleosts, BMP15 was first characterized in zebrafish [40] and has since been described in a few fish species, including Japanese flounder (*Paralichthys olivaceus*) [41], catfish (*Clarias batrachus*) [42], black porgy (*Acanthopagrus schlegelii*) [43], yellow-tail kingfish (*Seriola lalandi*) [44], European sea bass (*Dicentrarchus labrax*) [45], and gibel carp (*Carassius auratus gibelio*) [46]. Most of these studies have been limited to spatiotemporal expression patterns of *bmp15* without much exploration of its biological activities and functional importance. Incubation of zebrafish full-grown (FG) follicles with antiserum against zebrafish Bmp15 promoted oocyte maturation whereas treatment with recombinant human BMP15 suppressed human chorionic gonadotropin (hCG) and activin-stimulated oocyte maturation [40,47,48]. Knockdown and overexpression of *bmp15* in early zebrafish follicles suggested that Bmp15 might function to prevent premature oocyte maturation [47]. Overexpression of *bmp15* in a flounder ovarian cell line suppressed expression of steroidogenic genes including aromatase (*cyp19a1a*) [41]. In agreement with this, the expression of *bmp15* was negatively correlated with that of steroidogenic enzymes such as *cyp19a1a* and treatment of ovarian fragments with recombinant human BMP15 caused a decrease in the expression of *cyp19a1a* as well as 3β-HSD (*hsd3b*) and 17β-HSD (*hsd17b*) [42]. Disruption of *bmp15* gene in zebrafish resulted in follicle blockade at previtellogenic (PV) stage followed by sex reversal from females to males [49]. In contrast to the suppression of *cyp19a1a* expression by Bmp15 in other fish species, no signal of *cyp19a1a* expression could be detected in the granulosa cells of the zebrafish *bmp15* mutant [49]. Despite these studies, the exact mechanisms of BMP15 actions remain largely unknown.

To explore the mechanisms underlying BMP15 actions in controlling folliculogenesis, we undertook this genetic study in zebrafish. Zebrafish is an excellent model for studying folliculogenesis because the females spawn on daily basis with follicles developing continuously in the ovary [50]. We first created a *bmp15* mutant by CRISPR/Cas9. In contrast to the *gdf9* mutant (*gdf9-/-*) whose follicles were arrested at PG stage or PG-PV transition [22], the follicles in female *bmp15* mutant (*bmp15-/-*) were arrested at later stage, *i.e*., PV stage or PV-EV (early vitellogenic stage) transition, leading to female infertility. These results suggest that both Gdf9 and Bmp15 are critical in regulating early folliculogenesis; however, they act sequentially to control different stages. Our data also suggest that in addition to controlling the transition from PV to EV, Bmp15 also promotes follicle activation or puberty onset, which is marked by the first wave of PG-PV transition in fish ovary. Using both genetic and pharmacological approaches as well as transcriptome analysis, we provided insightful evidence for interactions of Bmp15 and the activin-inhibin system in regulating folliculogenesis, which involves

vitellogenin (Vtg/*vtg*) biosynthesis in the liver in response to estradiol (E2) produced by ovarian aromatase (*cyp19a1a*), and possibly Vtg uptake as well by growing oocytes via endocytosis through potential Vtg receptors (e.g., *lrp2a*).

## Results

### Spatiotemporal expression of *bmp15* and related genes in folliculogenesis

BMP15 was first discovered in mice and humans as an oocyte-specific growth factor [23]. However, a previous study using *in situ* hybridization showed that *bmp15* was expressed in both the oocyte and follicle cells in zebrafish [40]. We clarified this issue by mechanically separating oocyte and somatic follicle layer of FG follicles from adult female zebrafish at 5 months post-fertilization (mpf) according to our reported protocol [51]. This was followed by RT-PCR detection of *bmp15* expression in the two compartments. The well-known oocyte-specific *gdf9* and LH receptor (*lhcgr*) were used as the markers for denuded oocyte and follicle layer respectively. Both *gdf9* and *bmp15* mRNAs were detected exclusively in the denuded oocytes. In contrast, *lhcgr*, *inha* (inhibin α), *inhbaa* (inhibin/activin βAa), *bmpr2a* and *bmpr2b* (BMP type II receptors) were all expressed exclusively in the follicle layers, in agreement with our previous studies [52–54]. Such expression patterns strongly suggested a potential Bmp15-mediated paracrine pathway in the follicle that mediates an oocyte-to-follicle cell communication. We also examined the distribution of a few vitellogenin receptor-like proteins in the follicle, including *lrp1ab*, *lrp2a*, *lrp5* and *lrp6*, which were identified by transcriptome analysis (see below for details). Interestingly, *lrp1ab*, *lrp5* and *lrp6* were all expressed in both follicle cells and oocytes whereas *lrp2a* was exclusively expressed in the follicle cells (S1A Fig).

We also analyzed the temporal expression profiles of some of the above genes during folliculogenesis. Most of the genes examined showed the lowest expression at PG stage except *gdf9* and *bmp15*, which showed the highest expression at PG and PV stage respectively, followed by progressive decline towards FG stage (S1B Fig). The expression patterns of *fshr*, *lhcgr*, and *inha* all agreed well with our previous studies [52,55]. The expression patterns of *lrp1ab*, *lrp2a*, *lrp5* and *lrp6* seemed to correlate closely with the phase of vitellogenic growth, in particular *lrp2a*. The expression of *lrp2a* was nearly undetectable at previtellogenic PG and PV stages and post-vitellogenic LV (late vitellogenic) and FG stages; however, it surged dramatically at EV and mid-vitellogenic (MV) stage, which represent the stages of the fastest vitellogenic growth (S1B Fig).

### Establishment of *bmp15* knockout zebrafish line

To create *bmp15* null mutants, we designed a sgRNA that targets a site downstream of the translation start codon in the exon 1 of *bmp15* (S2A Fig). A mutant line was established with 5-bp indel deletion that introduced an early terminator, predicted to result in synthesis of a truncated protein (S2B Fig) (ZFIN line number: umo35). The indel mutation of *bmp15* was also confirmed at mRNA level by RT-PCR on the ovary using a specific primer with 3'-end overlapping with the mutation site, which would generate positive product in the wild type (WT, *bmp15*+/+) and heterozygotes (*bmp15*+/-), but not in the homozygous mutant (*bmp15*-/-) (S2C Fig). High-resolution melting analysis (HRMA) and heteroduplex mobility assay (HMA) were performed to identify the genotypes. Since the mutation is only 5-bp deletion, it is difficult to distinguish the homozygous mutant gene (*bmp15*-/-) from WT gene (*bmp15*+/+) by HRMA. To solve this problem, we spiked the unknown samples with WT DNA to form heteroduplexes with the DNA from the homozygous mutant, which would generate a melt curve similar to that of the heterozygote (S2D Fig). We performed HMA to

confirm the accuracy of HRMA. The heterozygotes showed two slow-migrating bands and the homozygous mutant showed a lower band than the WT (S2E Fig).

## Delayed follicle activation and puberty onset in *bmp15*-deficient females

To investigate the function of *bmp15* in early follicle development, we performed histological analysis of juvenile zebrafish females from 40 to 60 days post-fertilization (dpf) to study whether the lack of *bmp15* would affect puberty onset, which is marked by the appearance of the first cohort of PV follicles characterized with the formation of cortical alveoli [56] (Fig 1A). According to our recent studies, the PG follicles form from 25 to 35 dpf and remain at the same stage until about 45 dpf when some follicles start to enter the PV stage [57,58]. Upon entering the PV stage, the cortical alveoli appear first as a single layer of small vesicles in PV-I follicles (143±2 μm), then larger ones in PV-II follicles (180±11 μm) and finally multiple layers in PV-III follicles (254±22 μm) (Fig 1B). As we reported previously, the initiation of PG-PV transition or puberty onset in female zebrafish depends on body growth with standard body length (BL) of 1.8 cm and body weight (BW) of 100 mg being the thresholds [56,59]. This provides a valuable tool for assessing the regulation of puberty onset [60].

As shown in Fig 1A, the PV follicles started to appear in control females (*bmp15+/-*) at 45 dpf when BW and/or BL approached or crossed 100 mg and/or 1.80 cm, respectively, but not when they were both below the thresholds. In contrast, many mutant fish (*bmp15-/-*) could not undergo the PG-PV transition even when their BL and BW had both crossed the thresholds (1.80 cm and/or 100 mg). We measured BL and BW of three batches of fish at 45 dpf, with a total of 80 fish (18, 31 and 31 fish in each batch, respectively). The fish that had reached or exceeded the thresholds (1.8 cm and/or 100 mg) were genotyped and examined by histology. The female fish without PV follicles were considered to have delayed puberty onset and their percentage in all females examined was calculated. Statistical analysis showed that all the fish in the control group (*bmp15+/-*) had completed puberty onset after reaching the thresholds of BW and BL at 45 dpf; however, about 41% (33/80) mutants (*bmp15-/-*) were still at pre-pubertal stage without PV follicles (Fig 1C). The correlation between PG-PV transition and body size in *bmp15+/-* and *bmp15-/-* are shown in Fig 1D. Many mutant fish had PG follicles only even though their body size had exceeded the thresholds. PV follicles started to appear in the mutant ovary only when the BL and BW had reached 2.1 cm and 112 mg, respectively. The delayed PG-PV transition continued to be seen at 50 dpf (~29%, 2/7) (Fig 1E). In addition, the mutant fish that had initiated puberty onset had early PV follicles (PV-I or PV-II) only, while nearly all the control fish had reached late PV (PV-III) stage. As a result, the mutant ovaries contained more PG and less PV follicles (Fig 1F). This result clearly demonstrated that puberty onset or follicle activation was delayed in *bmp15*-deficient females.

## Post-pubertal arrest of follicle development in *bmp15*-deficient females

To investigate the function of *bmp15* in post-pubertal folliculogenesis, we performed histological examination of *bmp15* mutants at sexual maturation stage (120 dpf). The control fish (*bmp15+/+* and *bmp15+/-*) showed normal folliculogenesis with full range of follicles from PG to FG. In contrast, the follicles in the mutant (*bmp15-/-*) were completely blocked at the PV stage with normal formation of cortical alveoli but without accumulation of yolk granules in the oocytes (S3A Fig). Measurement of follicle diameters showed that none of the follicles in the mutant could grow beyond PV stage (S3B Fig). We further quantified follicle composition in the control (*bmp15+/-*) and mutant fish based on follicle size and structural features (cortical alveoli and yolk granules). In control fish, 30.1% of the follicles were at vitellogenic stage from EV to FG (stage III), 22.5% at PV stage (stage II), and 47.3% at PG stage (stage I). In

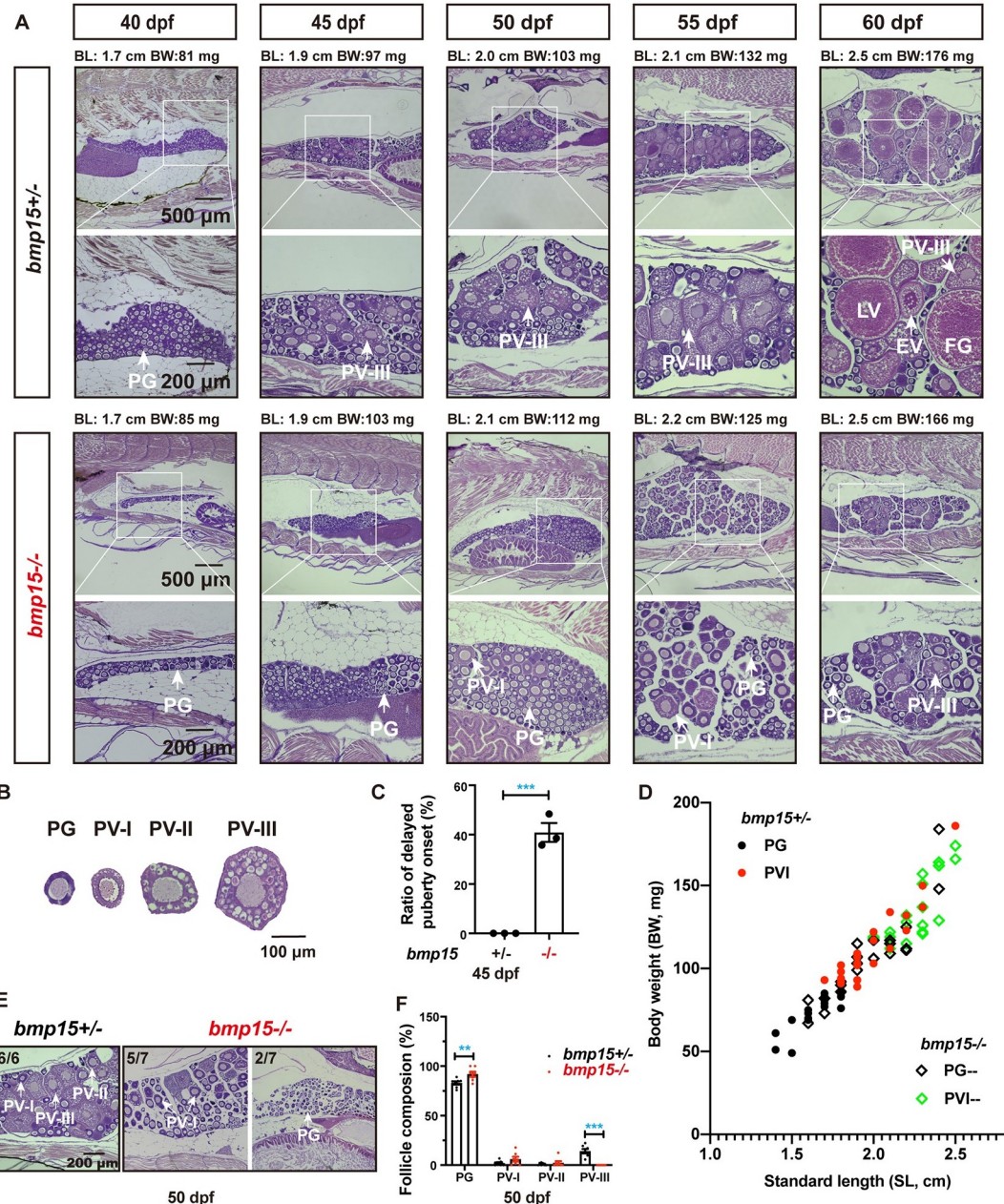

**Fig 1. Delayed follicle activation and puberty onset in *bmp15*-deficient females.** (A) Phenotype analysis of the early follicle development and PG–PV transition in control (*bmp15+/-*) and mutant (*bmp15-/-*) fish at prepubertal and pubertal stage (40–60 dpf). The boxed areas are shown at higher magnification below. The BL (cm) and BW (mg) of each fish are shown on the top. In control fish (*bmp15+/-*), PV follicles containing cortical alveoli started to appear when their BL and BW reached the threshold for puberty onset (1.80 cm and/or 100 mg). However, PV follicles did not appear in many *bmp15* mutant individuals although their body size had reached the threshold. (B) Classification of PV follicles. According to the size and number of layers of the cortical vesicle, the PV stage can be further divided into three sub-stages: PV-I, PV-II and PV-III. (C) Delayed puberty onset in the mutant (*bmp15-/-*) fish at 45 dpf (3 batches of samples with 18, 31 and 31 fish respectively). The female fish that reached or exceeded the thresholds (1.8 cm and/or 100 mg) but contained no PV follicle were considered to have delayed onset of puberty. The values are expressed as mean ± SEM and analyzed by t-test (*** P < 0.001; n = 3). (D) Correlation between PG-PV transition and body size [BL (cm) and BW (mg)]. Dots in different color represent different stages of follicles. The thresholds in control females were 1.8 cm and/or 100 mg while the thresholds in the mutant were 2.2 cm and/or 130 mg. (E) Histological analysis of the control and mutant fish at 50 dpf. All individuals (n = 6) in the control group had reached PV-III stage, while some of the mutants could only grow to PV-I and/or PV-II stage (5/7) and others remained in the PG stage (2/7). (F) Analysis of follicle composition at 50 dpf (n = 7). Compared with the control, the *bmp15-/-* ovary contained significantly more PG follicles but less PV-III follicles. The values are expressed as mean ± SEM and analyzed by ANOVA followed by the Tukey HSD for multiple comparisons (** P < 0.01; *** P < 0.001).

contrast, the mutant ovaries comprised 74.3% PG and 25.7% PV follicles, but no vitellogenic ones (S3C Fig).

## Sex reversal of *bmp15*-deficient females to males

To evaluate long-term effects of *bmp15* deficiency, we examined the mutant from 50 to 300 dpf. During this period, the testis and spermatogenesis were normal in mutant males. As for females, the mutant ovaries were largely normal at 80 dpf except that the follicles were arrested at PV stage with large interfollicular spaces. In addition to females (8/21) and males (10/21), we also observed ovotestis in some mutant fish (3/21), indicating sex reversal from females to males. At 210 dpf, most mutant fish were males (11/19) with some other fish at transitional state with ovotestis (6/19). The remaining mutant females (2/19) showed severe ovarian degeneration and follicle atresia (Fig 2A).

We also characterized the secondary sexual characteristics (SSCs) of zebrafish during the process of sex reversal and correlated them with gonadal conditions. In addition to the enlarged abdomen and silverish body color, the female zebrafish also possess a protruding cloaca or genital papilla (GP). The male zebrafish appear brownish in color with a slim body shape, and they develop unique breeding tubercles (BTs) on the pectoral fins [61]. The development of BTs has been shown to be androgen-dependent in zebrafish [62]. For the convenience of study, we divided the process of sex reversal into four stages (I-IV) based on changes of both gonads and SSCs. The fish at stage I has normal ovary as seen in control females with GP only. Stage II marks the start of sex reversal with the ovary containing follicles only without testicular tissues; however, while maintaining GP, the fish start to develop small BTs on the pectoral fins, an indication of masculinization. Stage II is therefore the period when both feminine GP and masculine BTs coexist in the same fish. As sex reversal progresses, the GP gradually regresses. The fish at stage III has lost GP while its BTs become more prominent. The ovary of stage III fish still contains follicles; however, the testicular tissues have appeared and become progressively dominant. Stage IV marks the end of sex reversal with a complete replacement of the ovary by testis as seen in control males. All these four stages could be seen in the mutants (*bmp15-/-*) at 120 dpf (Fig 2B). The development of BTs on the pectoral fins seemed to be a sensitive marker for sex reversal as its development precedes the appearance of visible testicular tissues in the ovary. Quantification of the BT area showed a clear correlation with the development of testicular tissues at different stages of sex reversal (Fig 2C). We also analyzed gonadosomatic index (GSI, gonad weight/body weight) of female, male and intersexual mutants at 120 dpf. The female mutants (*bmp15-/-*) at stage I showed significantly lower GSI than the control females, and the GSI of the intersexual mutants at stage II and III was even lower, close to that of males of both control and mutant (stage IV) (Fig 2D). Analysis of sex ratios from 50 to 300 dpf showed that the sex reversal process occurred at different times in different individuals starting from about 90 dpf and lasted beyond 300 dpf (Fig 2E).

## Partial rescue of *bmp15* deficiency by *inha* mutation

BMP15 and GDF9 are close members of the TGF-β superfamily, and they are both specifically expressed in the oocytes. Our recent study showed that the loss of *gdf9* in zebrafish resulted in a complete arrest of follicles at PG stage without accumulation of cortical alveoli. Interestingly, this phenotype could be partially rescued by simultaneous mutation of inhibin α subunit (*inha-/-*) [22]. This raised an interesting question about the relationship between Bmp15 and inhibin. To address this issue, we created a double mutant *bmp15-/-;inha-/-* using an *inha* mutant line established in our laboratory (ZFIN line number: umo19) and examined its folliculogenesis. According to our report, the *inha-/-* single mutant displayed precocious puberty

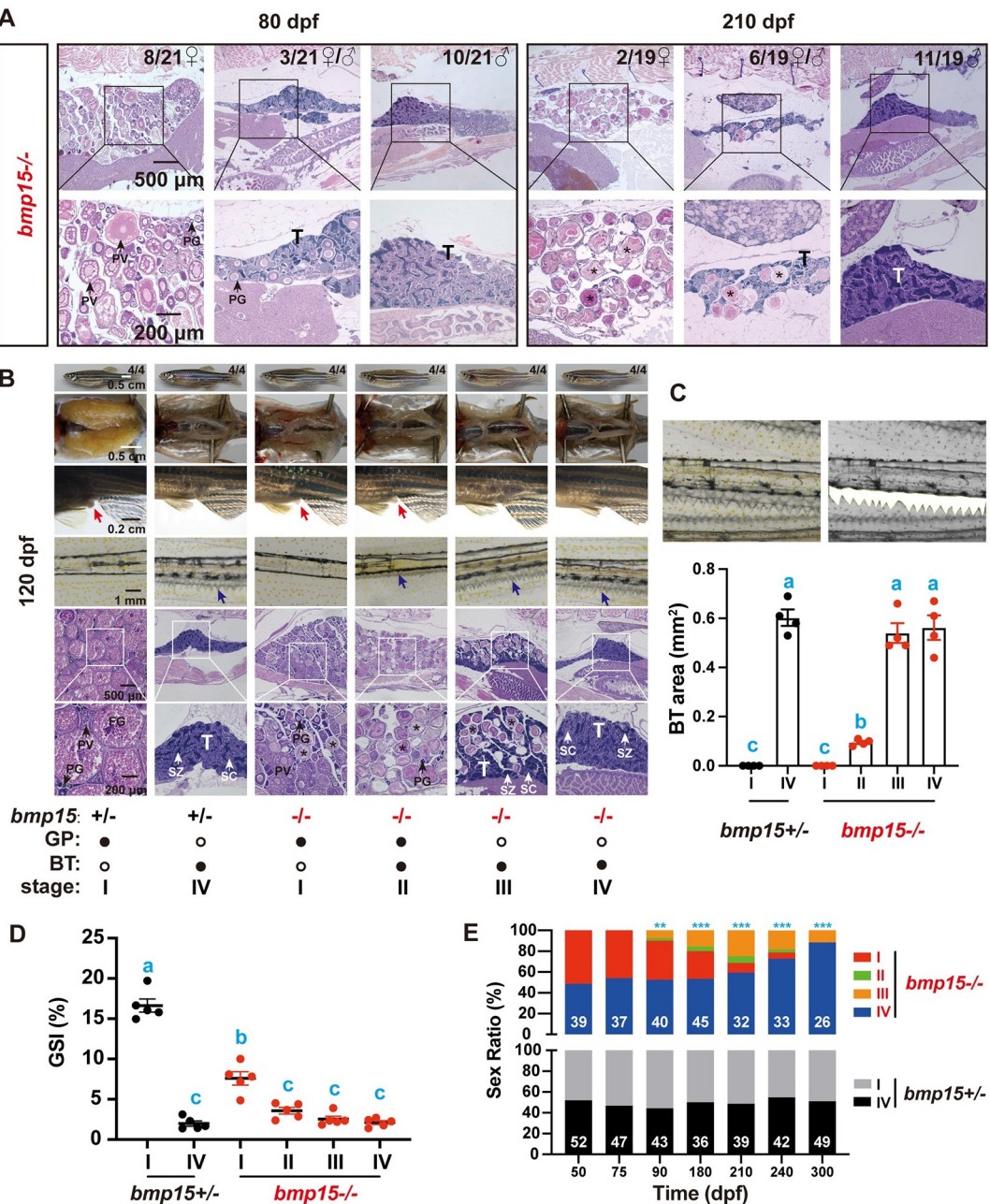

**Fig 2. Sex reverse in *bmp15*-deficient females.** (A) Histology analysis of the mutant (*bmp15-/-*) at 80 and 210 dpf. In addition to degenerating ovaries, individuals with ovotestes (♀/♂) were increasingly observed from 80 to 120 dpf, indicating a female-to-male sex reversal. Asterisk, degenerating oocytes; T, testicular tissue. (B) Morphology, gross anatomy, secondary sexual characteristics (GP and BTs) and gonadal histology. The genital papilla (GP, red arrow) was prominent at the cloaca in females while the breeding tubercles (BTs, blue arrows) were present on the pectoral fins of males. T, testicular tissue; SC, spermatocytes; SZ, spermatozoa. (C) Quantification of BT areas in the control and *bmp15* mutant undergoing different stages of sex reversal (stage I-IV). The white color marks the BT area for quantification. Stage I and IV represent female and male respectively whereas stage II and III represent transitional stages of sex reversal. The values are expressed as mean ± SEM (n = 4) and analyzed by ANOVA followed by Tukey HSD for multiple comparisons. Different letters indicate statistical significance (P < 0.05). (D) Gonadosomatic index (GSI, gonad weight/body weight) of the control (*bmp15+/-*) and mutant (*bmp15-/-*) fish at 120 dpf. The GSI of stage I mutant was significantly lower than in the control. The values are expressed as mean ± SEM (n = 5) and analyzed by ANOVA followed by the Tukey HSD for multiple comparisons. Different letters indicate statistical significance (P < 0.05). (E) Change of sex ratios during gonadal development in *bmp15+/-* and *bmp15-/-* fish from 50 to 300 dpf. The sex ratios in control fish were around 50:50 (♂: ♀) at all times examined; however, intersexuality started in

the mutant around 90 dpf with increasing males (stage IV). The data were analyzed by Chi-squared test compared with the control (** $P < 0.01$; *** $P < 0.001$).

onset or early PG-PV transition, but with normal follicle growth; however, its oocytes failed to mature and ovulate, leading to female infertility [63].

Interestingly, simultaneous mutation of *inha* also restored vitellogenic growth in *bmp15* mutant. In contrast to *bmp15-/-* single mutant whose follicles were arrested at PV stage with formation of cortical alveoli but not yolk granules in the oocytes, the double mutant *bmp15-/-; inha-/-* showed normal vitellogenic growth beyond PV stage with large amount of yolk accumulated in the oocytes (Fig 3A). However, quantitative analysis of follicle composition based on diameter showed that the partially rescued follicles could only grow to the MV stage, not LV and FG stage, suggesting a blockade at MV-LV transition (Fig 3B). Since yolk granules are derived from Vtg, which is produced in the liver in response to estrogens, we determined the levels of E2 and Vtg in the serum. The levels of E2 and Vtg decreased significantly in *bmp15-/-* single mutant; however, they were both restored to normal levels in the double mutant (*bmp15-/-;inha-/-*) (Fig 3C).

## Transcriptome analysis of *bmp15-/-*, *inha-/-* and *bmp15-/-;inha-/-* mutant follicles

To understand how mutation of *inha* could partially rescue the phenotypes of *bmp15* mutant, we performed a transcriptome analysis on PG follicles isolated from WT (*bmp15+/+;inha+/+*), single mutants (*bmp15-/-* and *inha-/-*) and double mutant (*bmp15-/-;inha-/-*). We chose PG follicles for the analysis because the *bmp15* mutant displayed a delay of follicle activation or PG-PV transition. The RNA-seq data revealed a total of 734 up-regulated and 1789 down-regulated genes in *bmp15-/-* mutant follicles compared with WT. In contrast, a total of 5572 genes were up-regulated and 3451 down-regulated in *inha-/-* mutant follicles. In the double mutant, 4852 genes were up-regulated and 3855 down-regulated, similar to that of *inha-/-* but distinct from that of *bmp15-/-*. The heatmap of differentially expressed genes (DEGs) shows clear and distinct patterns of the three genotypes (single mutants: *bmp15-/-* and *inha-/-*; double mutant: *bmp15-/-;inha-/-*) and the similarity between *inha-/-* and *bmp15-/-;inha-/-* (Fig 4A). Most of the DEGs in *bmp15-/-* follicles were down-regulated as compared to WT whereas most DEGs in *inha-/-* were up-regulated (Fig 4A and 4B). Interestingly, mutation of *inha-/-* in the double mutant reversed the expression pattern from *bmp15-/-* to *inha-/-*. Example genes include *ecm1a*, *rps4x*, *marco*, *nr1d2a* and *lrp2a*, which were down-regulated in *bmp15-/-* follicles but up-regulated in *inha-/-* and *bmp15-/-;inha-/-* follicles (Fig 4B). Such expression patterns suggested that Bmp15 may act as a facilitator of folliculogenesis, while inhibin acts as a depressant.

The DEGs with statistical significance were subjected to further GO enrichment and KEGG pathway analyses. Many GO terms associated with fundamental biological processes were enriched for up- and down-regulated genes in *bmp15-/-* mutants, including translation and cell respiration. In *inha-/-* and double mutants (*bmp15-/-;inha-/-*), the GO terms enriched for up- and down-regulated genes included RNA processing, immune response, defense response, cell migration and GTPase activity (S4 Fig). Interestingly, some genes belonging to the pathways of TGF-β signaling, endocytosis, and receptor-mediated endocytosis showed no change in expression in *bmp15-/-* (S5A Fig, for details see S1 Table) but increased expression in *inha-/-* (S5B Fig, for details see S2 Table) and *bmp15-/-;inha-/-* (S5C Fig, for details see S3 Table) compared to the control (*bmp15+/+;inha+/+*). The expression changes of all these

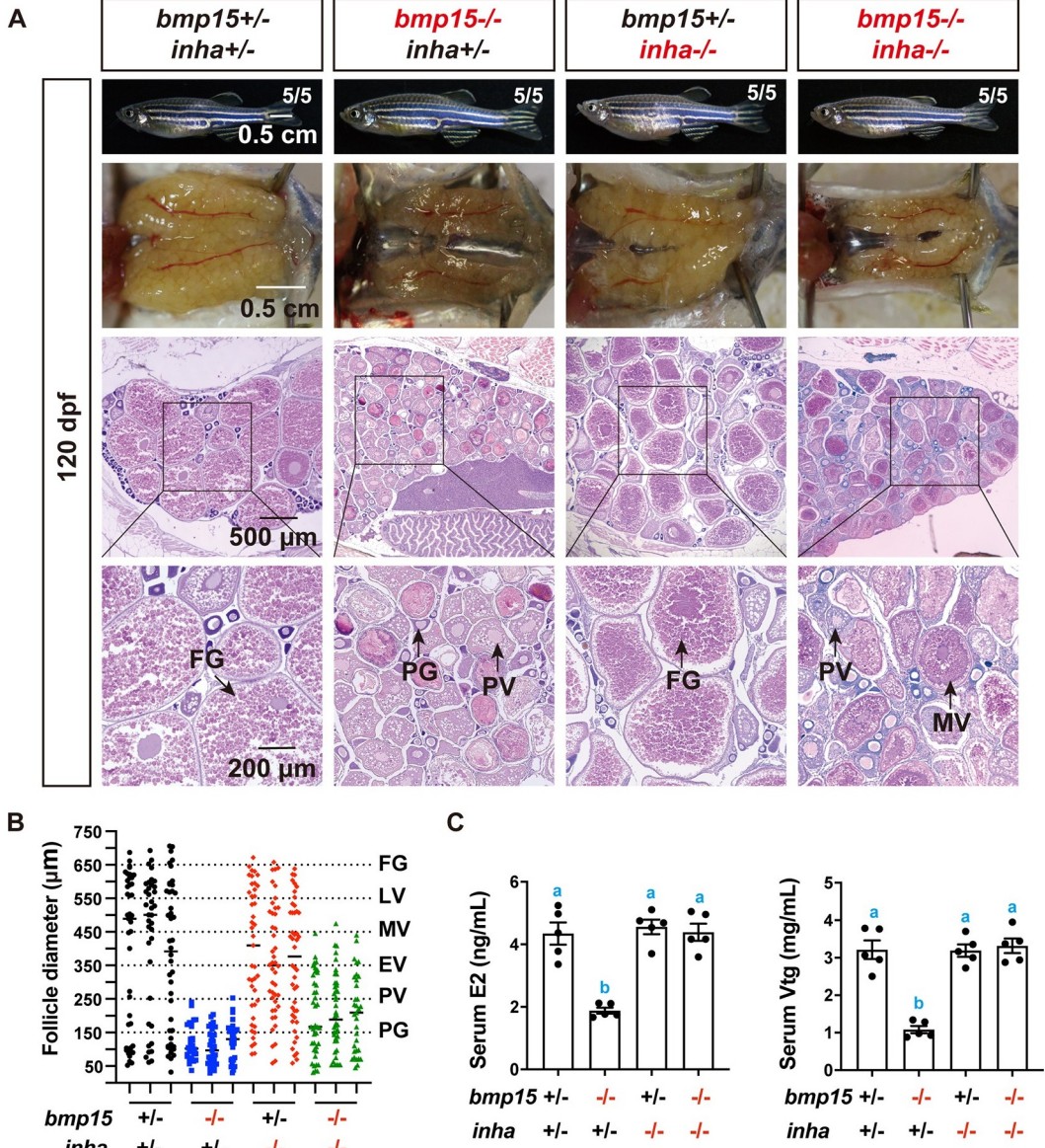

**Fig 3. Partial rescue of *bmp15* deficiency by *inha* mutation.** (A) Histological examination of gonads at 120 dpf. The follicle development was arrested at PV–EV transition in *bmp15-/-* females while the double mutant (*bmp15-/-;inha-/-*) resumed vitellogenic growth to MV stage, not FG stage as seen in *inha-/-*. (B) Follicle distribution in different genotypes at 120 dpf. The diameters of *bmp15* mutant follicles could reach the size of PV stage (~250 μm), while the double mutant (*bmp15-/-;inha-/-*) follicles could grow beyond PV to reach MV stage (~450 μm). (C) Serum levels of E2 and Vtg in different genotypes at 120 dpf. The E2 and Vtg levels were significantly lower in *bmp15-/-* females than those in the control, and they were both returned to normal levels in the double mutant (*bmp15-/-;inha-/-*). The values are expressed as mean ± SEM (n = 5) and analyzed by ANOVA followed by the Tukey HSD for multiple comparisons. Different letters indicate statistical significance (P < 0.05).

genes were verified by real-time qPCR (S5A–S5C Fig). In the TGF-β signaling pathway, activin subunit βAa (*inhbaa*), TGF-β1a (*tgfb1a*), BMP type II receptors (*bmpr2a* and *bmpr2b*) and two type I receptors of the TGF-β superfamily (*tgfbr1b* and *acvrl1*) showed significant increase in both *inha-/-* and *bmp15-/-;inha-/-*. The pathways of endocytosis and receptor-mediated endocytosis enriched for up-regulated genes in *bmp15-/-;inha-/-* were particularly interesting as

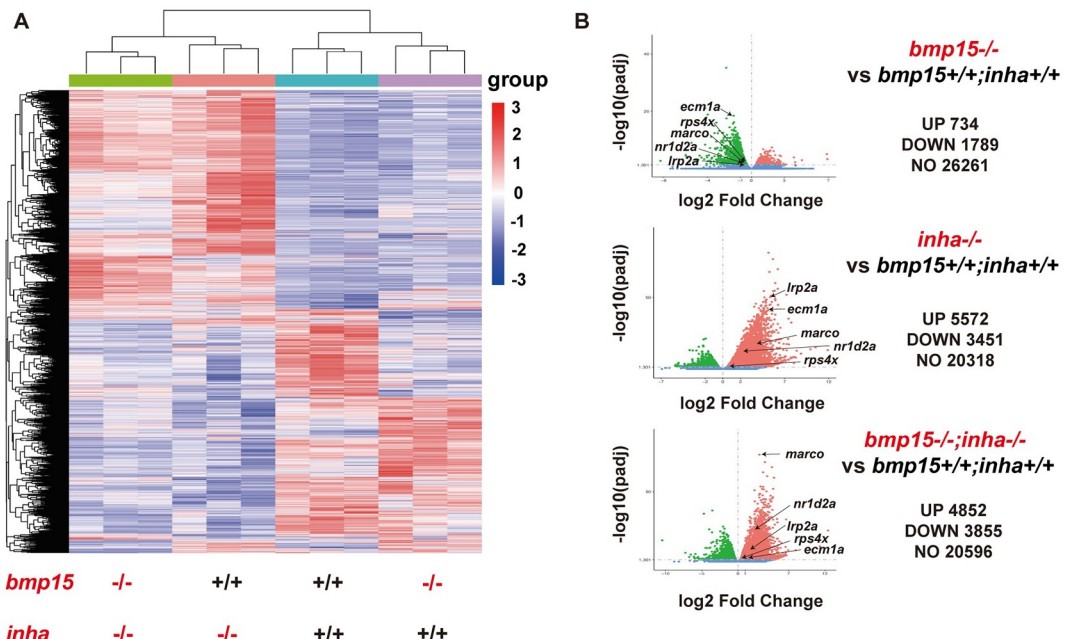

**Fig 4. Transcriptome analysis of *bmp15* and *inha*-deficient follicles.** (A) Heatmap of DGEs among four genotypes of *bmp15* and *inha* mutants. The heatmap of *bmp15-/-;inha-/-* showed similar pattern to that of *inha-/-* but not *bmp15-/-*. Regularized log transformed (rlog) count matrix was generated using DeSeq2 package and the DEGs with significance were used to plot the heatmap of rlog counts. (B) Volcano plot for DEGs of *bmp15-/-*, *inha-/-* and *bmp15-/-;inha-/-* compared with *bmp15+/+; inha+/+* respectively. Most DEGs in *bmp15-/-* follicles were down-regulated whereas most DEGs in *inha-/-* and *bmp15-/-; inha-/-* were up-regulated.

these genes might play roles in Vtg uptake, which involves receptor-mediated endocytosis. The increased expression and activity of these genes in *inha-/-* and *bmp15-/-;inha-/-* would suggest an enhanced Vtg uptake, resulting in yolk granule accumulation.

In zebrafish folliculogenesis, the PG-PV transition represents a critical stage of follicle development, which involves significant changes in gene expression [64,65]. Although our transcriptome data revealed dramatic changes in expression of thousands of genes in *bmp15-/-* and *bmp15-/-;inha-/-* at PG stage, it may not represent the changes at PV stage where the follicles of *bmp15-/-* were blocked. To demonstrate this, we performed real-time PCR at both PG and PV stages on some selected genes that are believed to play key roles in follicle development, including aromatase (*cyp19a1a*), gonadotropin receptors (*fshr* and *lhcgr*), and the activin-inhibin-follistatin system (*inhbaa*, *inhbab*, *inhbb*, *inha*, *fsta* and *fstb*). To ensure comparability, we isolated PG and PV follicles from the fish at 60 dpf when the WT and mutant fish had developed to the same stage (PV) (Fig 5A). Notably, the expression of *cyp19a1a* and *inhbaa* was very low at PG stage but increased sharply during the PG-PV transition in the control fish. The loss of *bmp15* did not significantly affect *cyp19a1a* and *inhbaa* expression in PG follicles, but dramatically reduced their expression in PV follicles. In contrast, none of the other genes examined showed significant response to *bmp15* mutation at either PG or PV stage (Fig 5B). To confirm this, we measured serum levels of E2 and Vtg by ELISA. As expected, both E2 and Vtg decreased significantly in *bmp15-/-* mutant females (Fig 5C). Since *cyp19a1a* is responsible for the production of E2, which is essential for hepatic production of Vtg, the reduced expression of *cyp19a1a* and production of Vtg could be one of major factors that prevented PV follicles from entering vitellogenic growth (EV-FG).

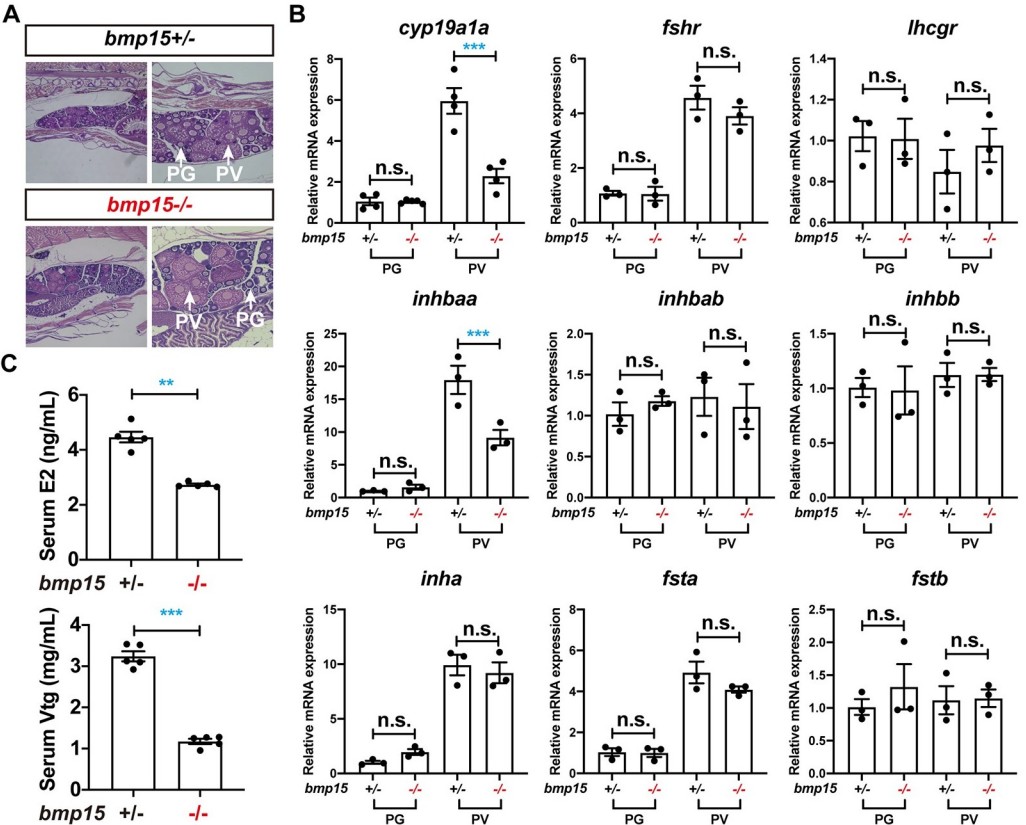

**Fig 5. Expression of selected genes at PG and PV stages in *bmp15* mutant.** (A) Stage-matching ovarian samples collected at 60 dpf. Both PG and PV follicles were present in *bmp15+/-* and *bmp15-/-* ovaries. (B) Expression of selected genes at PG and PV stages in *bmp15+/-* and *bmp15-/-* fish. The expression of *cyp19a1a* and *inhbaa* were dramatically reduced in PV follicles of *bmp15-/-* fish. *cyp19a1a*, ovarian aromatase; *fshr* and *lhcgr*, FSH and LH receptors; *inhbaa*, *inhbab* and *inhbb*, activin/inhibin β subunits; *inha*, inhibin α subunit; *fsta* and *fstb*, follistatins. The relative mRNA levels were determined by real-time qPCR, normalized to the housekeeping gene *ef1a*, and expressed as fold change compared with the levels at the PG stage of the control fish. The values are expressed as mean ± SEM (n = 5) and analyzed by ANOVA followed by the Tukey HSD for multiple comparisons (*** P < 0.001; n.s., no significance). (C) Levels of E2 and Vtg in the serum of *bmp15+/-* and *bmp15-/-* females at 120 dpf. The E2 and Vtg contents of *bmp15-/-* were both significantly lower than those in the control. The values are expressed as mean ± SEM (n = 5) and analyzed by t-test (** P < 0.01; *** P < 0.001).

## Resumption of vitellogenic growth in *bmp15* mutant females by estrogens

To investigate if the blockade of folliculogenesis at PV stage in *bmp15-/-* females was due to reduced production of estrogens as shown above. We performed an experiment *in vivo* to test if exposure to E2 could rescue the follicle blockade in *bmp15-/-* females. Two methods were used for the treatment: water-borne exposure and oral administration by feeding. The fish were treated for 20 days from 80 to 100 dpf. For water-borne exposure, E2 was first dissolved in ethanol and added to the tank to the final concentration of 10 nM. For oral administration, the fish were fed with E2-containing diet twice a day at 5% (W/W) of total fish body weight each time. Both feeding at 200 μg/g and water-borne exposure at 10 nM suppressed ovarian development, in agreement with our previous report on the toxic effect of E2 on folliculogenesis at high dosage [66]. However, E2 at lower doses of 2 and 20 μg/g were effective in promoting ovarian growth in *bmp15-/-* females. The dose of

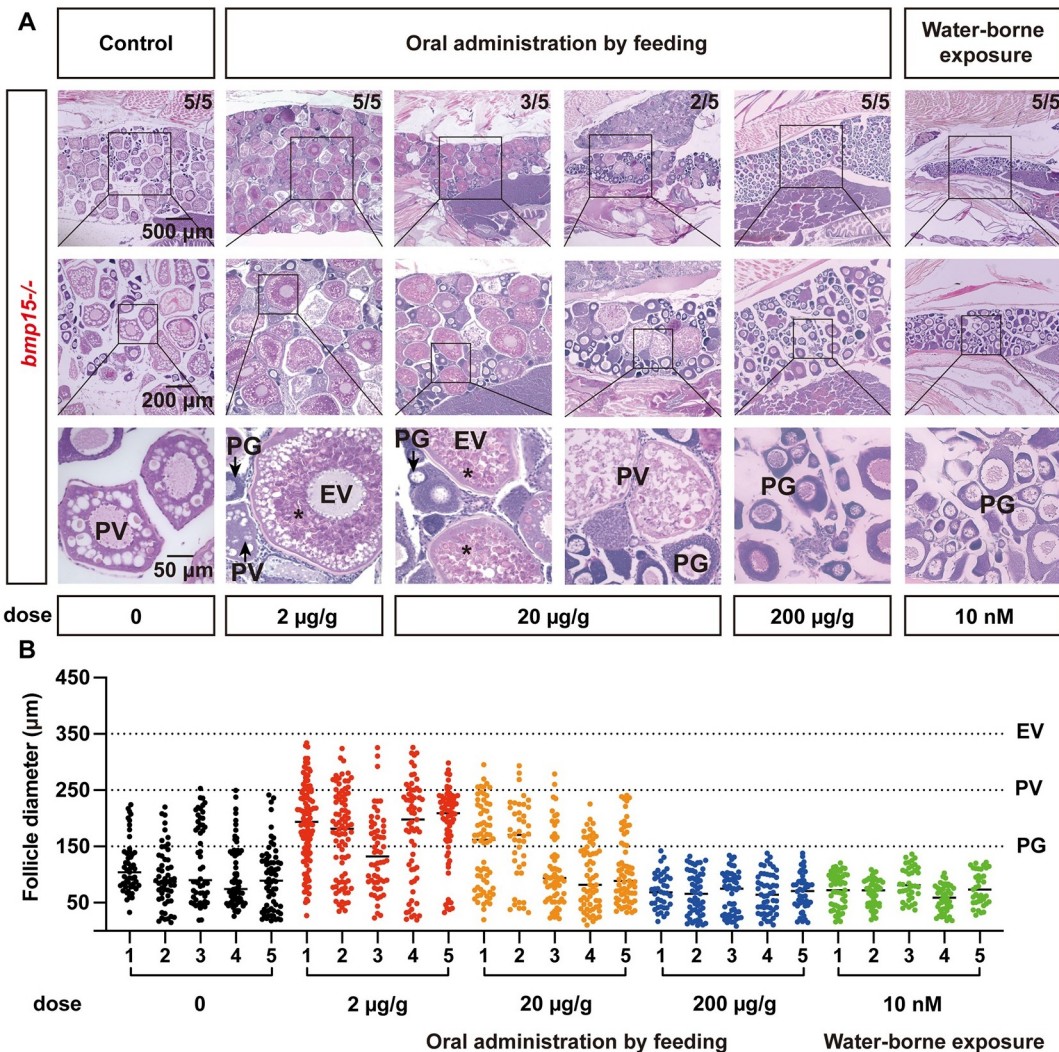

**Fig 6. Rescue of vitellogenic growth in *bmp15-/-* females by E2 treatments.** (A) Histology analysis of the *bmp15-/-* ovary after E2 treatment. The mutant fish were treated from 80 to 100 dpf by E2 via water-borne exposure (10 nM) or oral administration by feeding (2, 20 and 200 μg/g diet). All five fish fed with E2-containing diet at 2 μg/g resumed vitellogenic growth with yolk mass (asterisk) whereas some fish at 20 μg/g (3/5) contained vitellogenic follicles. Both water-borne exposure and feeding at 200 μg/g suppressed follicle activation or PG-PV transition. The numbers shown in each sample indicate total number of fish sampled (lower) and the number of fish that showed the same phenotype (upper). (B) Follicle distribution in *bmp15-/-* ovary after E2 treatments. The mutants could break the blockade at PV stage after treatment with E2 at 2 and 20 μg/g diet and their follicles could enter vitellogenic growth to reach the size range of EV stage (~ 350 μm).

2 μg/g showed the strongest and most consistent effect as all five fish treated overcame the blockade at the PV stage and resumed vitellogenic growth (5/5) compared to the group of 20 μg/g (Fig 6A). However, measurement of follicle sizes showed that although E2 treatment could resume vitellogenic growth with yolk accumulation in the mutant, the follicle could only grow to the EV stage (Fig 6B), suggesting that estrogens were likely one of the key factors involved in the partial rescue of *bmp15-/-* phenotypes by *inha-/-*, but not the only factor.

## Involvement of *cyp19a1a* in the partial rescue of *bmp15* deficiency by *inha* mutation

As described above, both *inha* mutation (*inha-/-*) and E2 treatment could partially rescue the phenotypes of *bmp15-/-*, *i.e.*, resumption of follicle development beyond PV stage with accumulation of yolk granules. Since the expression of *cyp19a1a* decreased significantly in *bmp15-/-* follicles (Fig 5B) but increased dramatically in *inha-/-* [63], it is conceivable that the partial rescue of *bmp15-/-* phenotype by *inha-/-* might involve *cyp19a1a* and E2; in other words, Bmp15 may act in zebrafish follicles by increasing *cyp19a1a* expression and therefore E2 production. To test this hypothesis, we treated the double mutant fish *bmp15-/-;inha-/-* with fadrozole, an aromatase (Cyp19a1a) inhibitor, by oral administration from 80 to 100 dpf. The result showed that the resumption of vitellogenic growth in *bmp15-/-;inha-/-* was completely abolished by treatment with fadrozole (200 μg/g) (Fig 7A). Gene expression analysis in both PG and PV follicles and determination of serum E2 and Vtg levels further confirmed the effectiveness of fadrozole. The expression of *cyp19a1a* was low in PG and PV follicles of both control *bmp15+/-* and mutant *bmp15-/-* fish; however, it increased significantly in both PG and PV follicles from E2-treated *bmp15-/-* as well as *bmp15-/-;inha-/-* fish. Fadrozole treatment abolished the increase in the double mutant (Fig 7B). Similar response was also observed for Vtg genes *vtg1* and *vtg3* in the liver (Fig 7C). In agreement with the changes in gene expression, the serum E2 and Vtg levels were both low in *bmp15-/-* fish, and E2 treatment and *inha* mutation (*bmp15-/-;inha-/-*) both increased their concentrations to the control levels. Such increase in the double mutant was again abolished by fadrozole treatment (Fig 7D).

## Involvement of *inhbaa* in the partial rescue of *bmp15* deficiency by *inha* mutation

In addition to *cyp19a1a*, our transcriptome analysis also demonstrated up-regulation of TGF-β signaling pathway in both *inha-/-* and *bmp15-/-;inha-/-*, including two ligands (*tgfb1a* and *inhbaa*) and several type II and I receptors (*bmpr2a*, *bmpr2b*, *tgfbr1b* and *acvrl1*) (S5 Fig). Among these genes, *inhbaa* was particularly interesting as its expression decreased significantly in the PV follicles of *bmp15-/-* fish, in contrast to other subunits of the activin-inhibin-follistatin system (Fig 5B). As inhibin (αβ) is the antagonist of activin (ββ), the loss of *inha* (α) and increased expression of *inhbaa* (βAa) would suggest an increase in activin formation and its activity. We therefore postulated that in addition to *cyp19a1a*, activins derived from Inhbaa might also play a role in the phenotypical rescue of *bmp15-/-* by *inha* deficiency among other TGF-β family members mentioned above. To test this hypothesis, we created a triple mutant using an *inhbaa* mutant line established in our laboratory (ZFIN line number: umo27) [67]. The *inhbaa-/-* single mutant showed progressive deterioration of the ovary with reduced fecundity after 6 mpf, and its folliculogenesis stopped completely with follicles arrested at PG stage after 18 mpf. Despite ovarian deterioration, the *inhbaa-/-* mutant was able to maintain its female characteristics without undergoing sexual reversal [67]. Phenotype analysis at 120 dpf showed that the lack of *inhbaa* in the triple mutant (*bmp15-/-;inha-/-;inhbaa-/-*) prevented most of the phenotypes rescued by *inha-/-* in *bmp15-/-;inha-/-*. First, although the follicles in the triple mutant continued to grow beyond the size range of PV stage, they could not undergo normal vitellogenic growth as no yolk granules were visible in the cytosol of the oocytes between the germinal vesicle (GV) and the zone of cortical alveoli, in contrast to that seen in the double mutant (Fig 8A and 8B). As the result, the triple mutant ovary looked lighter in color with follicles barely visible with naked eyes, similar to that of *bmp15-/-* (Fig 8A). Second, *inhbaa* deficiency also retarded follicle growth in the triple mutant compared to that in the double mutant. The follicles in the triple mutant could grow close to the size of EV stage (< 350 μm), but not

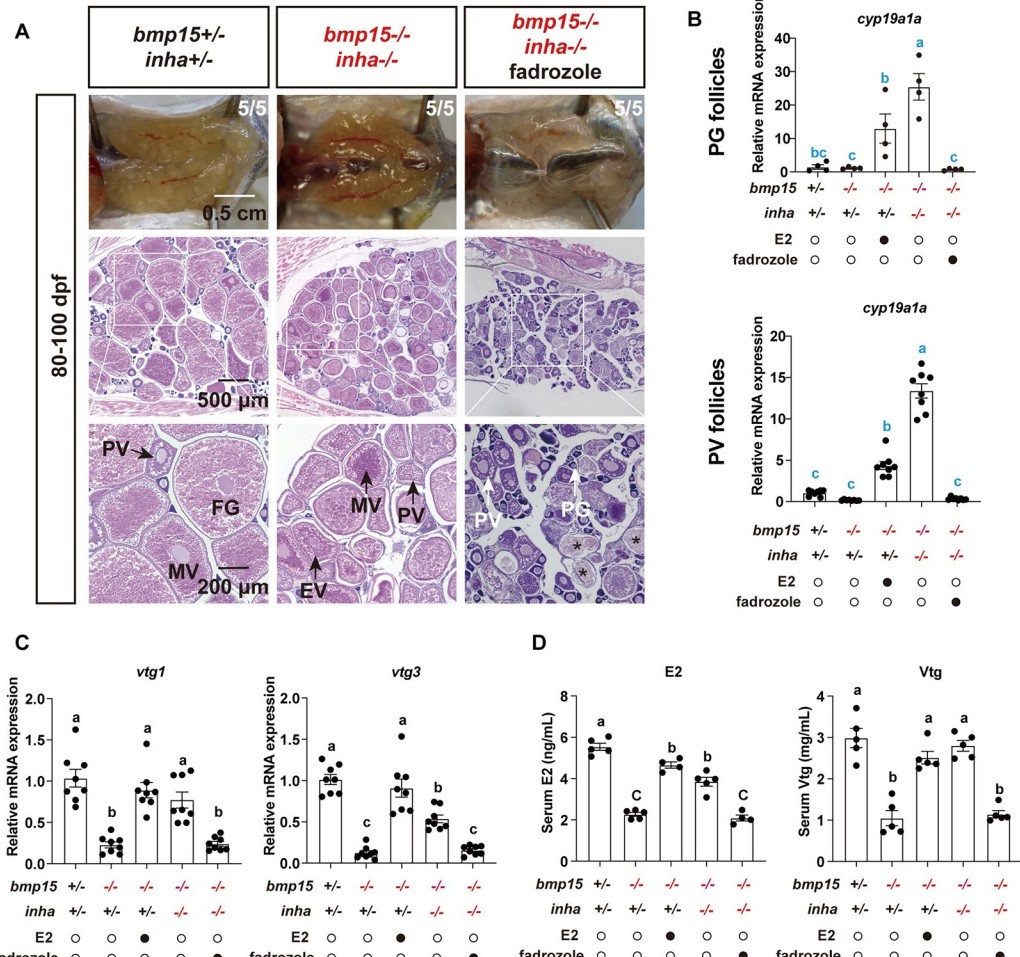

**Fig 7. Effect of fadrozole on vitellogenic growth in *bmp15* and *inha* double mutant.** (A) Morphology and histology of the double mutant gonads (*bmp15-/-;inha-/-*) after fadrozole treatment. The females were reared in 10 L tanks from 80 to 100 dpf with dried powder feed containing fadrozole (200 μg/g diet) at 10% (W/W) of fish body weight per day. The resumption of follicle development to vitellogenic stage in the double mutants was completely abolished by treatment with fadrozole. The numbers shown in each sample indicate total number of fish sampled (lower) and the number of fish that showed the same phenotype (upper). (B) Expression of *cyp19a1a* in the PG and PV follicles of the control and mutants after E2 and fadrozole treatments. (C) Expression of *vtg1* and *vtg3* in the liver of the control and mutants after E2 and fadrozole treatments at 120 dpf. The relative mRNA levels were determined by real-time qPCR, normalized to the housekeeping gene *ef1a*, and expressed as fold change compared with the levels in the control fish. (D) Levels of E2 and Vtg in the serum of the control and mutants after E2 and fadrozole treatments. The values are expressed as mean ± SEM (n ≥ 3) and analyzed by ANOVA followed by the Tukey HSD for multiple comparisons. Different letters indicate statistical significance (P < 0.05).

MV stage (~450 μm) as achieved in the double mutant (Fig 8C). As the result, the GSI in the triple mutant was similar to that in *bmp15-/-* single mutant but significantly lower than that in the double mutant (Fig 8D). Third, both E2 and Vtg levels in the serum decreased significantly in the triple mutant compared to those in the double mutant (Fig 8E and 8F). Due to the deficiency in follicle growth, all single (*bmp15-/-*; PV), double (*bmp15-/-;inha-/-*; MV) and triple mutant (*bmp15-/-;inha-/-;inhbaa-/-*; EV) females were infertile as their follicles could not grow to the FG stage (Fig 8G). Interestingly, being arrested at PV stage, the *bmp15-/-* females eventually underwent sex reversal at different time points of development. However, the femaleness was maintained, and sex change did not occur in either double or triple mutants probably because

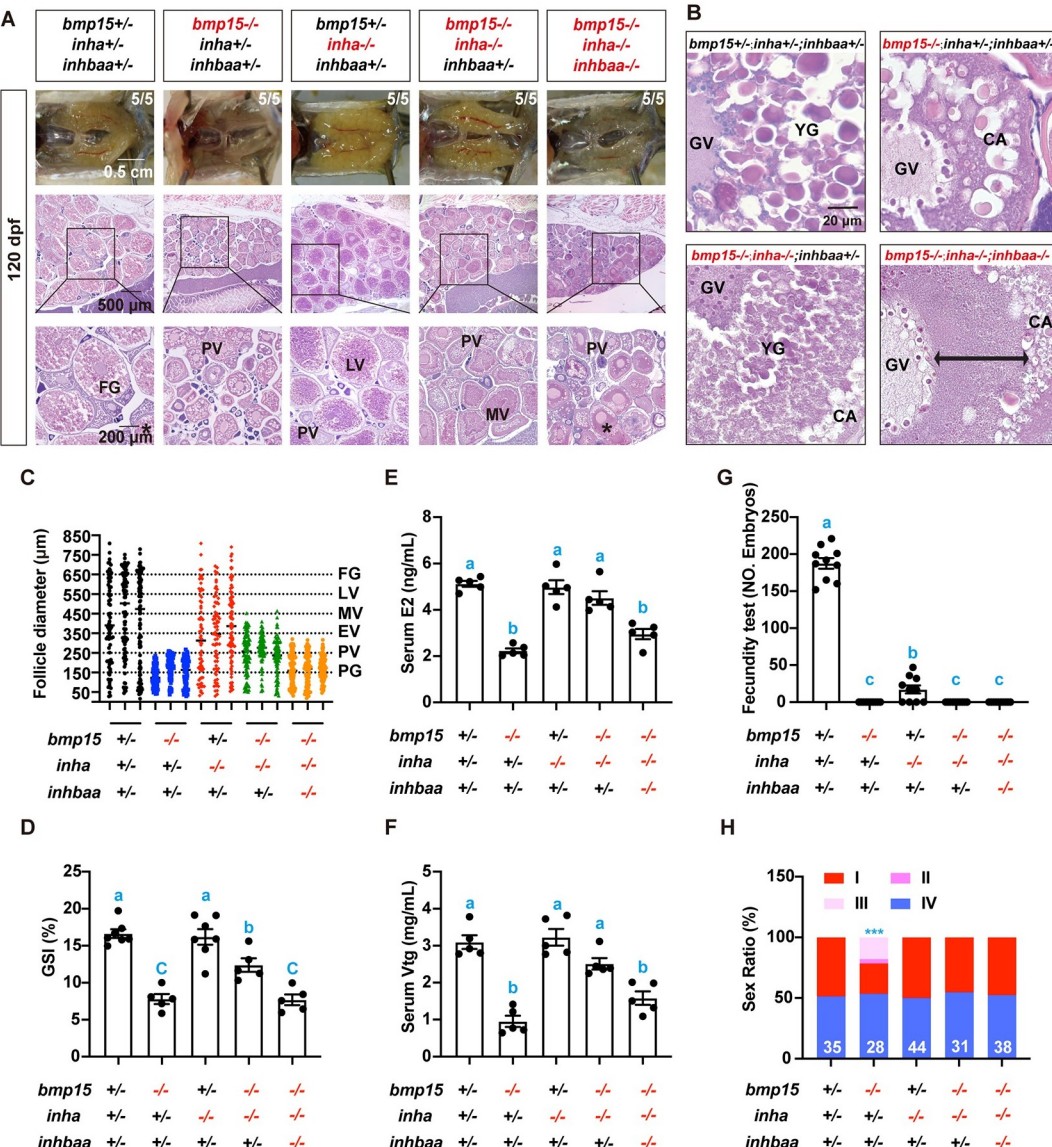

**Fig 8. Role of *inhbaa* in the partial rescue of vitellogenic growth in *bmp15*-/- by *inha*-/-.** (A) Morphology and histology of the gonads in different genotypes at 120 dpf. The asterisk shows vitellogenic follicles without yolk mass in the triple mutant (*bmp15*-/-;*inha*-/-;*inhbaa*-/-). (B) Oocytes from diffident genotypes at higher magnification. The oocytes in *bmp15*-/- single mutant were arrested at PV stage with cortical alveoli (CA) but no yolk granules (YG). The double mutant with *inha*-/- showed normal vitellogenic growth beyond PV stage with both CA and YG, while the lack of *inhbaa* in the triple mutant blocked YG accumulation again in the oocytes. The double arrowed bar shows the region of clear cytosol between the germinal vesicle (GV) and CA zone where the YG is supposed to be located. (C) Follicle distribution in the ovary of diffident genotypes. The follicles could grow to the size range of PV (~250 μm), MV (~450 μm) and EV stage (~350 μm) in the single (*bmp15*-/-), double (*bmp15*-/-;*inha*-/-) and triple mutant (*bmp15*-/-;*inha*-/-;*inhbaa*-/-), respectively. (D) GSI of females in different genotypes at 120 dpf. The values are expressed as mean ± SEM (n ≥ 5) and analyzed by ANOVA followed by the Tukey HSD for multiple comparisons. Different letters indicate statistical significance (P < 0.05). (E and F) Serum levels of E2 and Vtg in different genotypes at 120 dpf (n = 5). The normal concentrations of E2 and Vtg in the double mutants decreased again by the loss of *inhbaa* in the triple mutants. The values are expressed as mean ± SEM and analyzed by ANOVA followed by the Tukey HSD for multiple comparisons. Different letters indicate statistical significance (P < 0.05). (G) Fertility test of different genotypes at 120 dpf. Five female fish from each type were tested 10 times with normal WT males by natural breeding. The number of fertilized eggs was counted and analyzed after each breeding. The *inha* mutant females were sub-fertile with much lower fecundity than the control, while other genotypes of the mutant females were all infertile. (H) Sex ratios of different genotypes at 120 dpf. The double mutant and triple mutant females could maintain sexuality with sex ratio being around 50:50 (♂: ♀), whereas sex reversal occurred in *bmp15*-/- fish. I-IV, stages of sex reversal; *** P < 0.001 by Chi-squared test compared with the control.

the follicles become less prone to deterioration and sex reversal after PV stage (Fig 8H). Interestingly, the loss of *inhbaa* did not affect the phenotypes of *bmp15-/-* and *inha-/-* as the double mutants *bmp15-/-;inhbaa-/-* and *inha-/-;inhbaa-/-* displayed similar phenotypes to those of *bmp15-/-* and *inha-/-* single mutants, respectively, at 120 dpf (S6 Fig).

## Potential interactions of Bmp15, activin and inhibin

As described above, the defective phenotypes of *bmp15-/-* could be partially rescued by the loss of inhibin (*inha-/-*) in *bmp15-/-;inha-/-*. Furthermore, the phenotypical rescue of *bmp15-/-* by *inha-/-* could be partially prevented by the loss of activin βAa subunit (*inhbaa-/-*) in the triple mutant (*bmp15-/-;inha-/-;inhbaa-/-*). To explore the mechanisms by which Bmp15, inhibin (*inha*) and activin (*inhbaa*) might work together to regulate early follicle development, we performed RT-PCR analysis at 120 dpf on expression levels of some well-known factors that are likely involved in the event, *i.e.*, the transition from previtellogenic to vitellogenic growth (PV-EV transition), including gonadotropins in the pituitary (*fshb* and *lhb*) and their receptors in the ovary (*fshr* and *lhcgr*), aromatase (*cyp19a1a*) in the ovary and estrogen receptors in the liver (*esr1*, *esr2a* and *esr2b*), Vtg genes in the liver (*vtg1*, *vtg2*, *vtg3*, *vtg4*, *vtg5*, *vtg6* and *vtg7*) and Vtg receptor-like proteins in the ovary (*lrp1ab* and *lrp2a*).

In the ovary, we used PG follicles for the analysis as these follicles could be isolated more uniformly to minimize the interference by the follicles of advanced stages. Most of the genes examined were significantly up-regulated in the follicles of *inha-/-* mutant and remained higher in the double mutant (*bmp15-/-;inha-/-*) than the control fish. Among the ovarian genes examined, the expression of *cyp19a1a* showed the most dramatic changes in response to *bmp15*, *inha* and *inhbaa* mutations. Its expression was very low in both control and *bmp15-/-* follicles; however, it surged in *inha-/-* mutant and remained high in the double mutant (*bmp15-/-;inha-/-*), together with *inhbaa*. Interestingly, this increased expression was abolished in the triple mutant with *inhbaa* deficiency (*bmp15-/-;inha-/-;inhbaa-/-*). Similar pattern was also observed for *inhbaa*, *lrp1ab* and *lrp2a*. As for *fshr*, although its expression was also high in *inha-/-* mutant, it was not statistically significant (Fig 9A).

In the liver, the Vtg genes (*vtg1*, *vtg2*, *vtg3*, *vtg4*, *vtg5*, *vtg6* and *vtg7*) all decreased in expression in *bmp15-/-* mutant but significantly increased in *inha-/-* mutant despite that not all changes showed statistical significance. Although the expression of these Vtg genes decreased in the double mutant (*bmp15-/-;inha-/-*) compared to *inha-/-*, most of them showed higher expression than *bmp15-/-* mutant despite lack of statistical significance. Such difference was diminished again in the triple mutant without *inhbaa* (Fig 9B). Since Vtg production in the zebrafish liver is tightly controlled by estrogens from the ovary [68], we also examined expression patterns of estrogen receptors in the liver in various mutants. Among the three nuclear estrogen receptors (*esr1*, *esr2a* and *esr2b*), *esr1* displayed the most significant changes in expression, similar to the *vtg* genes. Its expression increased dramatically in *inha-/-*, but decreased sharply in *bmp15-/-;inha-/-* and *bmp15-/-;inha-/-;inhbaa-/-*. In contrast to *esr1*, the expression of *esr2a* and *esr2b* increased in all types of mutants, but there was no statistical significance compared with controls (Fig 9C).

At the pituitary level, *lhb* showed no statistically significant response to mutations of *bmp15*, *inha* and *inhbaa*. In contrast, the expression of *fshb* increased progressively from single mutants (*bmp15-/-* and *inha-/-*) to double (*bmp15-/-;inha-/-*) and triple (*bmp15-/-;inha-/-; inhbaa-/-*) mutants. Interestingly, instead of reducing or abolishing the increased expression in *inha-/-* or *bmp15-/-;inha-/-* as seen for other genes examined in the ovary and liver, the loss of *inhbaa* in the triple mutant further increased the expression of *fshb* in the pituitary (Fig 9D). This was also confirmed by *in situ* hybridization for *fshb* and *lhb*. The pituitary of the triple

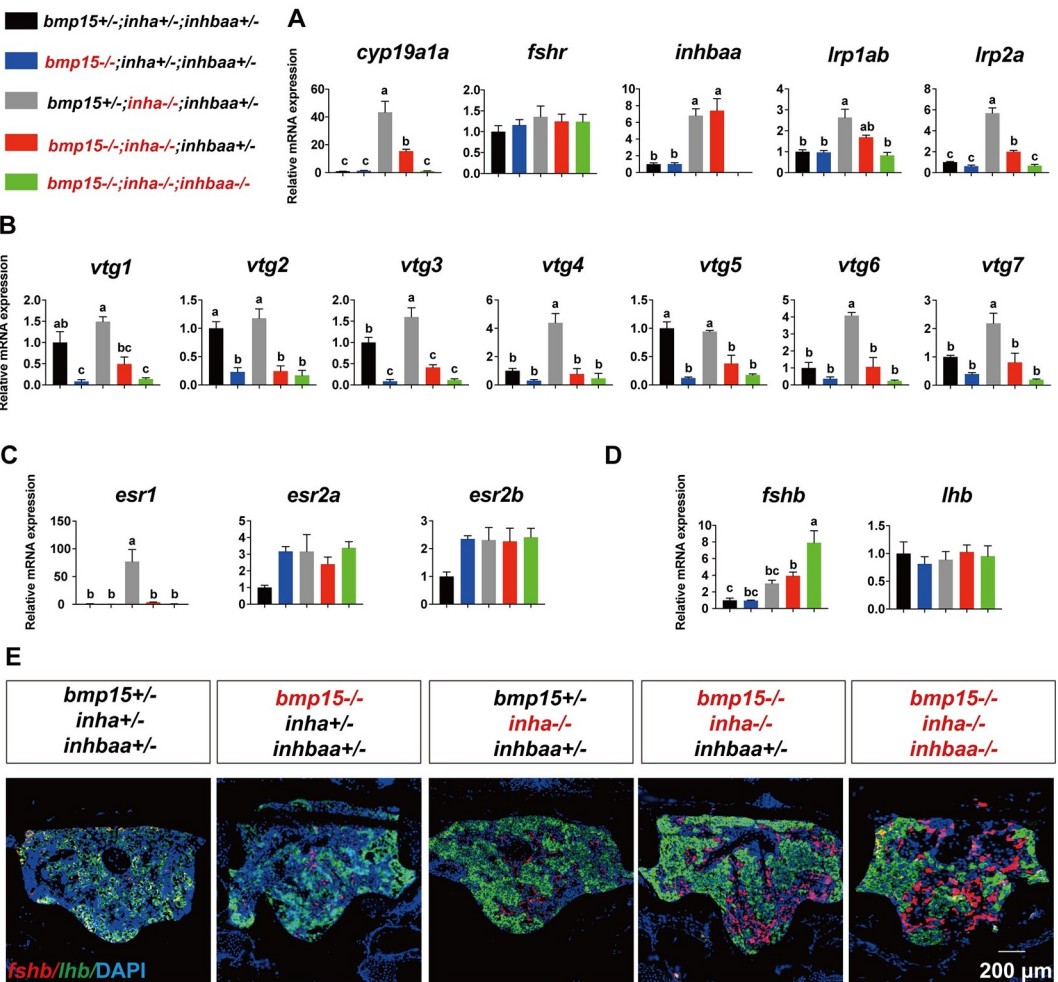

**Fig 9. Expression of selected genes in different genotypes at 120 dpf.** (A) Expression of ovarian genes in the PG follicles (*cyp19a1a*, *fshr*, *lrp1aa* and *lrp2a*). (B) Expression of Vtg genes in the liver (*vtg1-7*). (C) Expression of estrogen receptors in the liver (*esr1*, *esr2a* and *esr2b*). (D) Expression of gonadotropins in the pituitary (*fshb* and *lhb*). The relative mRNA levels were determined by real-time qPCR, normalized to the housekeeping gene *ef1a* and presented as the fold change compared with the control fish. Data were analyzed by ANOVA followed by the Tukey HSD for multiple comparisons (* P < 0.05; ** P < 0.01; *** P < 0.001; n ≥ 3). (E) *In situ* hybridization on the expression of *fshb* (red) and *lhb* (green) in the pituitary of different genotypes. DAPI stains for nuclei (blue).

mutant showed the strongest signal for *fshb* in the triple mutant (Fig 9E). The further increase in *fshb* expression in the triple mutant could be due to the loss of both inhibition by inhibin and reduced negative feedback by estrogens.

## Interaction of Bmp15, Gdf9 and Inhibin in controlling folliculogenesis

Bmp15 and Gdf9 are both oocyte-specific factors and they are closely related in structure and function. Interestingly, the loss of *bmp15* and *gdf9* resulted in follicle blockade at different developmental stages in zebrafish. The mutation of *gdf9* caused a complete arrest of follicle development at PG stage without formation of cortical alveoli (PG-PV transition) [22] whereas *bmp15* deficiency arrested follicles at PV stage (PV-EV transition) without any signs of vitellogenesis characterized by yolk granule accumulation. To study potential interactions between

*bmp15* and *gdf9*, we generated a double mutant (*bmp15-/-;gdf9-/-*) using a *gdf9* mutant we recently reported (ZFIN line number: umo18) [22]. As expected, the double mutant showed similar phenotype to that of *gdf9-/-* mutant at 90 dpf with most follicles arrested at PG stage; however, some follicles in most females (9/13) could break the blockade at PG stage to enter early PV stage (PV-I), more advanced that those observed in *gdf9-/-* single mutant but not further to PV-II and III as seen in *bmp15-/-* (Fig 10A).

Since *inha* deficiency could partially rescue the follicle blockade in both *gdf9* [22] and *bmp15* mutants, we also created a triple mutant (*bmp15-/-;gdf9-/-;inha-/-*) to see if *inha* mutation had any impact on the double mutant of *bmp15* and *gdf9*. The results showed that *inha* mutation could also help the double mutant (*bmp15-/-;gdf9-/-*) to overcome the follicle blockade to enter vitellogenic growth with formation of both cortical alveoli and yolk granules (Fig 10B). However, the vitellogenic follicles were mostly at EV stage without reaching the size range of MV stage (EV-MV transition) as seen in *bmp15-/-;inha-/-* and *gdf9-/-;inha-/-* (Fig 10C). Follicle composition analysis showed many fewer vitellogenic follicles in the ovary of the triple mutants (*bmp15-/-;gdf9-/-;inha-/-*) compared to the double mutants (*bmp15-/-;inha -/-* or *gdf9-/-;inha -/-*). The ratios of EV-FG follicles (stage III) in *bmp15-/-;inha -/-* and *gdf9-/-; inha -/-* were 54.7% and 55.0% respectively while the ratio dropped significantly to only 22.6% in the triple mutant (Fig 10D), suggesting impaired vitellogenic growth of follicles in the triple knockout.

## Normal spermatogenesis in *bmp15* mutant

At both anatomical and histological levels, the disruption of *bmp15* had no impact on testis development and spermatogenesis as observed at 180 dpf. No difference was observed either in the double mutant of *bmp15* and *inha* (*bmp15-/-;inha-/-*) or triple mutant with *inhbaa* (*bmp15-/-;inha-/-;inhbaa-/-*) compared with the control fish (S7 Fig), similar to our observations in *inha* and *inhbaa* single mutants [63,67].

## Discussion

Folliculogenesis is a highly coordinated developmental process, and the active role played by oocytes has been an attractive issue for research in the past twenty years [69]. GDF9 and BMP15 are the first and best characterized oocyte-derived growth factors that play essential roles in regulating follicle growth and maturation [70]. In contrast to GDF9 that has been demonstrated to be essential for early follicle development and therefore female fertility in both mice and zebrafish by loss-of-function studies [19,22], BMP15 shows significant functional variation among different mammalian species especially between mono-ovulatory (*e.g.*, sheep) and poly-ovulatory (*e.g.*, mice) species [71,72]. The loss of BMP15 in sheep leads to female infertility with underdeveloped ovaries containing primary follicles only [33,35–37]; however, its loss in mice does not generate significant impact on ovarian development and reproductive performance [32]. In zebrafish, females deficient in *bmp15* were arrested at the PV stage followed by sex reversal to become fertile males as observed in the present study and reported previously [49]. Despite these studies, the exact functions of BMP15 remain elusive and its action mechanisms are largely unknown. In this study, we performed extensive genetic analysis on functions of Bmp15 in folliculogenesis from puberty onset to fertility. More importantly, by generating a series of double and triple mutants with other genes including *inha*, *inhbaa*, and *gdf9*, we obtained novel clues to the mechanisms by which Bmp15 may work in controlling zebrafish folliculogenesis. Our major discoveries are discussed below.

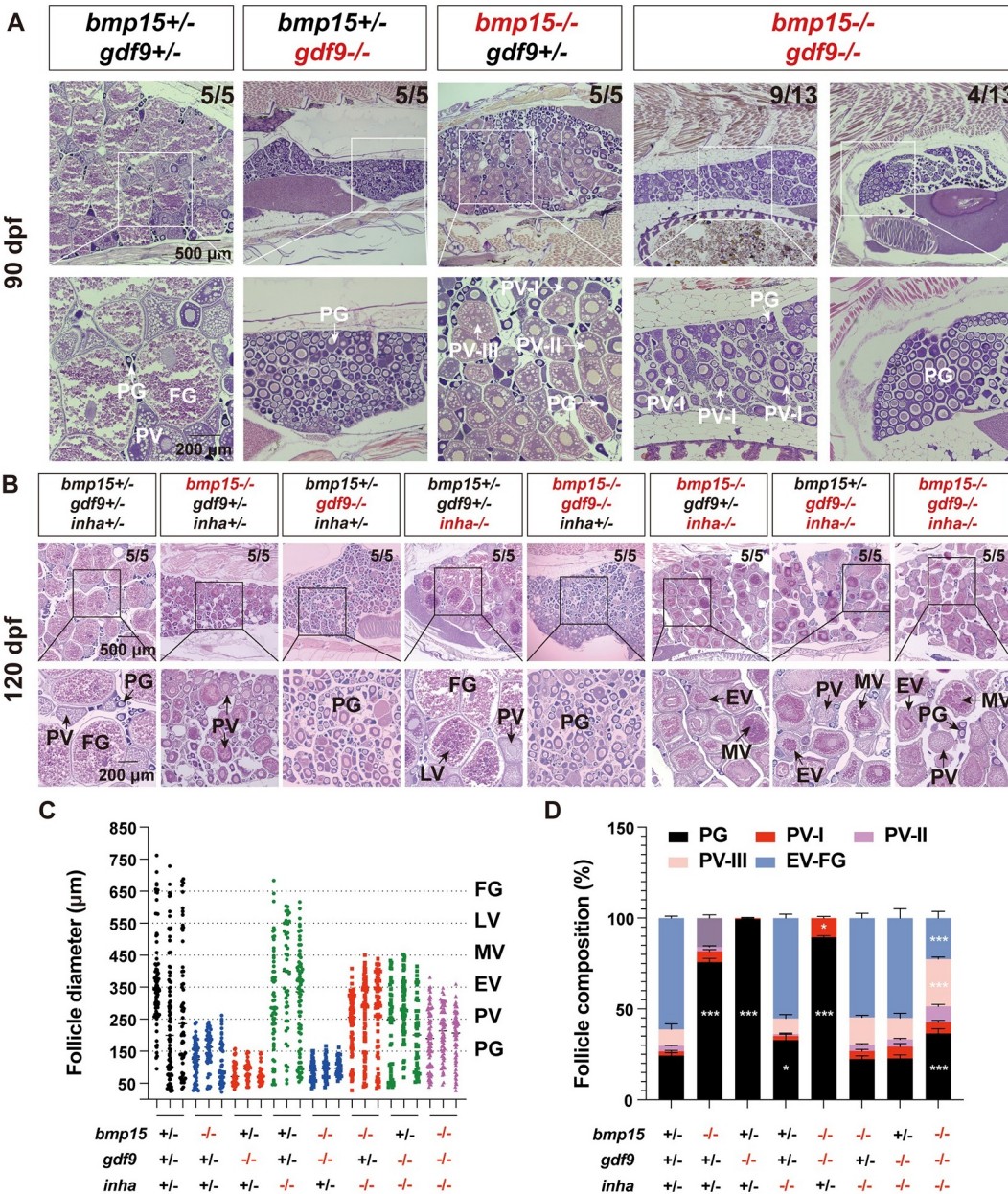

**Fig 10. Interaction of *bmp15* and *gdf9* in zebrafish follicle development.** (A) Histological analysis of *bmp15* and *gdf9* single and double mutants at 90 dpf. The follicles of some double mutants (*gdf9-/-;bmp15-/-*) could overcome the blockade at PG to enter early PV (PV-I) stage. (B) Partial rescue of vitellogenic growth in *bmp15* and *gdf9* single and double mutants by *inha-/-*. (C) Follicle distribution in different genotypes. Simultaneous mutation of *inha-/-* could rescue the follicle growth in both *bmp15-/-* and *gdf9-/-* to MV stage; however, further loss of *inhbaa* reduced it to EV stage. (D) Follicle composition in different genotypes and the data are shown as mean ± SEM (n = 3). Statistical analysis was performed with ANOVA followed by Tukey HSD for comparison with corresponding stages in the control fish (* $P < 0.05$; ** $P < 0.01$; *** $P < 0.001$). The proportion of PV-I follicles was significantly higher in the double mutant (*gdf9-/-;bmp15-/-*) than others especially *gdf9-/-* single mutant. The proportion of follicles at EV-FG stage (stage III) was lower in the triple mutants than the double mutant (*gdf9-/-;bmp15-/-*).

## Role of Bmp15 in female puberty onset

Puberty onset is a major event in reproductive development and maturation. Despite extensive studies, the mechanisms that control puberty onset remain elusive. Our studies in the past few years have established zebrafish as an excellent model for studying puberty onset [4,56,59,60,63]. In females, the appearance of the first wave of PV follicles in the developing ovary or PG-PV transition has been used as the marker for puberty onset [56]. We have demonstrated that body growth is a key factor in controlling puberty onset in zebrafish with 1.8 cm in BL and 100 mg in BW being the threshold for triggering the first PG-PV transition or puberty onset [56,59,63]. Our recent work on *gdf9* in zebrafish showed that the loss of *gdf9* gene caused a complete arrest of follicle development at PG stage, leading to failed puberty onset and therefore female infertility [22], similar to that in mice [19].

Despite its close relationship with *gdf9*, the loss of *bmp15* gene (also called GDF9B) in zebrafish did not cause a cessation of follicle development at PG stage as seen in *gdf9* mutant. The follicles could develop beyond PG and grow to the full size of PV stage with accumulation of cortical alveoli. However, the mutant fish (*bmp15-/-*) showed a significant delay in PG-PV transition or puberty onset. Many fish that had grown beyond the somatic threshold (1.8 cm/100 mg) remained at PG stage. This observation suggests that Gdf9 and Bmp15 in zebrafish play differential roles in controlling follicle activation or PG-PV transition with Gdf9 acting as an essential factor and Bmp15 being a promoting one (Fig 11A and 11B).

## Blockade of folliculogenesis in *bmp15*-deficient females

In contrast to GDF9 whose loss resulted in a complete arrest of follicle development at early stage in both mice (primary follicle) and zebrafish (primary growth) [19,22], the impacts of BMP15 mutation vary in different mammalian species. The *Bmp15* null mice did not show significant abnormalities in female reproduction. Other than reduced ovulation and fertilization, the female mutant mice had normal ovary and folliculogenesis [32]. However, the homozygous BMP15 (*fecX*) mutant in sheep exhibited abnormal follicle development from fetal to adult stage. Although follicles could form in the mutant, they failed to develop beyond the primary stage with abnormal arrangement of somatic follicle cells [36,37].

In agreement with a previous study [49], we also found that the loss of *bmp15* in zebrafish resulted in a complete arrest of follicle development at the PV stage characterized with normal formation of cortical alveoli but not yolk granules in the oocyte, indicating defects in vitellogenesis. Therefore, both Gdf9 and Bmp15 play important roles in controlling early follicle development in zebrafish; however, they act sequentially at different stages of folliculogenesis, *i.e.*, PG-PV and PV-EV transitions, respectively. Gdf9 is primarily involved in controlling PG-PV transition or follicle activation as a determinant [22], whereas Bmp15 controls subsequent PV-EV transition, the beginning of vitellogenic growth (Fig 11B).

## Roles of *bmp15* in maintaining ovarian function and femaleness

Although our data showed that Bmp15 was an oocyte-specific growth factor as in other species, it did not seem to affect gonadal differentiation as shown by the normal sex ratio of the young *bmp15-/-* fish. However, the ratio started to change after about three months (~90 dpf) with decreasing females and increasing males, and the mutant fish became nearly all males at 300 dpf (Fig 11A). This is similar to the phenotype of *gdf9* mutant, which also underwent sex reversal from about 60 dpf to become all males after about four months (~120 dpf) [22]. This phenomenon also occurred in some other mutants of zebrafish. For example, the null mutant of FSH receptor (*fshr-/-*) had underdeveloped ovaries with follicles being blocked at early PG stage followed by sex reversal to males [73]. In the triple mutant of estrogen receptors (*esr1-/-;*

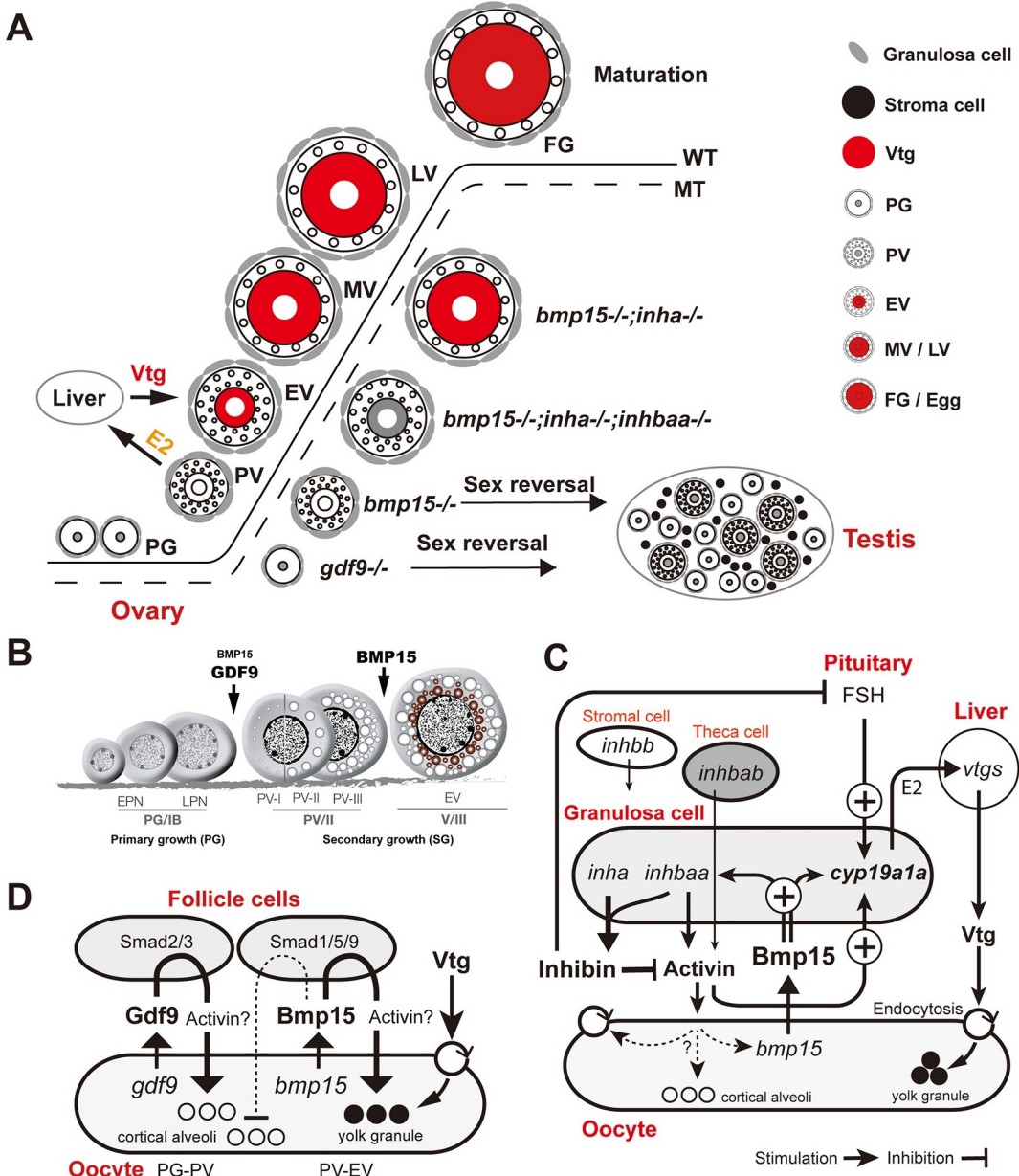

**Fig 11. Summary of genetic analysis and working hypotheses on mechanisms of Bmp15 action in zebrafish.** (A) Zebrafish folliculogenesis and phenotypic defects of single (*bmp15-/-* and *gdf9-/-*), double (*bmp15-/-;inha-/-*) and triple (*bmp15-/-; inha-/-;inhbaa-/-*) mutants. The loss of *gdf9* and *Bmp15* caused a complete arrest of follicle development at PG and PV stage, respectively. The *bmp15-/-* follicles showed normal formation of cortical alveoli but no yolk mass in the oocyte, followed by sex reversal to males. Double mutation with *inha* (*bmp15-/-;inha-/-*) prevented sex reversal and partially rescued the vitellogenic growth with yolk mass to MV stage. Further knockout of *inhbaa* in the triple mutant (*bmp15-/-;inha-/-;inhbaa-/-*) resulted in the loss of yolk granules again but allowed the oocytes to grow to EV size. (B) Roles of Gdf9 and Bmp15 in controlling early follicle development in zebrafish. Gdf9 is primarily involved in controlling PG-PV transition or follicle activation as a determinant; in contrast, Bmp15 is a key factor controlling PV-EV transition, the subsequent stage that marks the start of vitellogenic growth. Bmp15 is also involved in promoting PG-PV transition as an accelerator. EPN, early perinucleolar; LPN, late perinucleolar. (C) Actions and interactions of Bmp15 from the oocyte, activin/inhibin from the follicle cells and FSH from the pituitary in controlling aromatase (*cyp19a1a*) expression and Vtg biosynthesis as well as uptake. (D) Potential roles for Bmp15 and Gdf9 in regulating formation of cortical alveoli and yolk granules. Gdf9 may stimulate the biogenesis of cortical alveoli in the oocyte, while Bmp15 may play an antagonistic role in this regard while promoting yolk formation.

*esr2a-/-;esr2b-/-*), gonadal differentiation occurred normally and the follicles could develop to PV stage with formation of cortical alveoli; however, all follicles regressed and the mutant ovaries all changed to testes at later stage [74]. The mutation of epidermal growth factor receptor (*egfra-/-*) also resulted in a complete blockade of follicle development at PG or early PV stage followed by sex reversal to males, similar to that of *gdf9* mutant [75].

Interestingly, during the process of sex reversal in *bmp15* mutant, the external female and male secondary sex characteristics GP and BTs showed dynamic changes. The BTs seemed to be a sensitive marker for the onset of sex reversal as they appeared even before any testicular tissues were visible in the ovary and therefore coexisted with the GP for a short period of time. The mechanisms that triggered sex reversal in *bmp15* mutant are unknown. Reduced estrogen levels could be one of the factors involved as estrogens are key hormones stimulating Vtg production and therefore yolk formation in the oocytes [76] and the loss of estrogen receptors resulted in sex reversal as well [74]. We further explored this mechanism in the present study.

## Partial rescue of *bmp15* null phenotypes by simultaneous loss of *inha* - a novel clue to the mechanism of Bmp15 actions

Our recent study showed that the loss of *gdf9* in zebrafish resulted in a complete arrest of follicle development at the PG stage [22]. Surprisingly, double mutation of *gdf9* and *inha* could partially rescue the phenotypes of *gdf9-/-* to a great extent, including sex reversal, formation of cortical alveoli (PG-PV transition) and accumulation of yolk granules (vitellogenic growth) [22]. Such rescue is specific because *inha-/-* could not rescue the same phenotype of *egfra-/-*, i.e., cessation of follicle development at PG stage [75]. Surprisingly, simultaneous mutation of *bmp15* and *inha* also partially rescued the phenotypes of *bmp15-/-* females, similar to that seen in *gdf9-/-* [22], suggesting shared signaling and downstream mechanisms for Gdf9 and Bmp15. The follicles of *bmp15-/-;inha-/-* females could develop beyond the PV stage with accumulation of yolk granules. This provided a valuable tool to explore the mechanisms underlying Bmp15 actions in controlling early follicle development.

Transcriptome analysis showed clearly that the expression pattern of *bmp15-/-* was reversed by *inha-/-* in the double mutant. Among the biological processes and pathways identified by GO and KEGG analyses on DEGs, we were particularly interested in the TGF-β signaling pathway, endocytosis and receptor-mediated endocytosis. Many genes of these pathways, including *inhbaa*, were up-regulated in *inha-/-* and *bmp15-/-;inha-/-* follicles. However, these genes did not exhibit a significant decrease in expression in *bmp15-/-* compared with the control, as would be expected. This may be attributed to the fact that we used PG follicles for the transcriptome analysis. During this stage, before follicle activation and the transition to PV stage, many genes were expressed at very low levels, making it challenging to observe significant reductions in expression. Supporting this notion is the expression patterns of *cyp19a1a* and *inhbaa*, as well as their responses to *bmp15* mutation. These genes were expressed at low levels in PG stage and did not show any response to *bmp15* loss (*bmp15-/-*) as shown by real-time qPCR analysis. However, both genes exhibited a significant increase in expression during PG-PV transition, and the loss of *bmp15* caused a substantial decrease in their expression at PV stage. The loss of *inha* resulted in a remarkable increase in *cyp19a1a* expression, potentially due to the increased expression of *inhbaa* and subsequent enhanced production and activity of activin.

TGF-β family is well known to be important in reproductive development and function [14,77] whereas endocytosis especially receptor-mediated endocytosis is involved in the uptake of Vtg by the growing oocytes [78]. This together with the RT-PCR data on expression of specific functional genes such as *cyp19a1a* and *vtg1-7* and measurement of serum E2 and Vtg

levels has led us to hypothesize that the blockade of follicles at PV stage without yolk granule formation in *bmp15-/-* females might be due to deficiencies in both Vtg biosynthesis in the liver and its uptake by growing oocytes in the ovary.

## Roles of estrogens in Bmp15 regulation of vitellogenin synthesis

Estrogens are the major endocrine hormones that stimulate Vtg biosynthesis in the liver of non-mammalian oviparous vertebrates, therefore playing fundamental roles in vitellogenesis [79]. The lack of yolk granule accumulation or failure of vitellogenic growth in *bmp15* mutant zebrafish prompted us to speculate on possible involvement of estrogens in Bmp15 actions. Although the transcriptome analysis on PG follicles did not reveal any expression change of ovarian aromatase (*cyp19a1a*), the enzyme that catalyzes synthesis of estrogens from androgens, RT-PCR analysis on PV follicles demonstrated a significant drop of *cyp19a1a* expression in *bmp15* mutant, suggesting a stimulatory role for Bmp15 in regulating aromatase expression and therefore estrogen secretion and Vtg biosynthesis. This agrees well with a recent report in zebrafish that no *cyp19a1a* expression could be detected by *in situ* hybridization in the ovary of *bmp15* mutant [49]. The stimulation of aromatase expression by BMP15 has also been reported in mammals. Co-treatment of human cumulus granulosa cells with BMP15 and GDF9 greatly enhanced aromatase expression and estradiol secretion in response to FSH probably via Smad2/3 pathway [80]. In agreement with this, the blood estradiol level in the sheep carrying homozygous mutations of *BMP15* was undetectable [36].

Interestingly, the decreased expression of *cyp19a1a* in *bmp15-/-* females was rescued and raised to higher levels by simultaneous mutation of *inha* in the double mutant (*bmp15-/-; inha-/-*), accompanied by restored E2 level in the serum, increased Vtg production in the liver, and resumed vitellogenic growth of oocytes with accumulation of yolk granules. However, the follicle growth could only proceed to the MV stage, not LV and FG stage, suggesting a defect of certain regulatory mechanisms at MV stage that prevented MV-LV transition. This will be an interesting issue to explore in future studies. In addition to rescuing vitellogenesis, the loss of *inha* also prevented sex reversal in *bmp15-/-*, probably due to the increased estrogen production again.

To further confirm that the lack of vitellogenesis in *bmp15* mutant was due to the reduced expression of *cyp19a1a*, we performed two critical experiments: treatment of *bmp15-/-* females with E2 and the double mutant (*bmp15-/-;inha-/-*) with aromatase inhibitor fadrozole. E2 treatment restored vitellogenic growth with yolk granule accumulation in *bmp15-/-* follicles and treatment with fadrozole prevented the rescue of the *bmp15-/-* phenotype by *inha-/-*. These results, together with reduced serum levels of E2 and Vtg in *bmp15-/-* females, strongly suggest that one major function of Bmp15 in the zebrafish follicles is to stimulate *cyp19a1a* expression, therefore increasing estrogen production and hepatic Vtg biosynthesis (Fig 11C).

## Roles of the activin-inhibin system in Bmp15 regulation of vitellogenic growth

One of the interesting findings from our transcriptome analysis was the significant enrichment of up-regulated genes in the TGF-β signaling pathway in the double mutant PG follicles (*bmp15-/-;inha-/-*) compared to the single mutant (*bmp15-/-*), suggesting potential roles for TGF-β superfamily members in overcoming the *bmp15-/-* phenotypes. Among the members up-regulated, activin βAa subunit (*inhbaa*) was particularly interesting as it was also significantly up-regulated in the double mutant of *gdf9* and *inha* (*gdf9-/-;inha-/-*), which also rescued the deficiency of *gdf9-/-* [22]. The facts that Gdf9 and Bmp15 are closely related oocyte-specific growth factors, and their deficiencies could both be partially rescued by the simultaneous loss

of *inha*, accompanied by a significant increase in *inhbaa* expression, raise an interesting question about roles of activin and inhibin in Bmp15 function.

Activin and inhibin are both dimeric proteins consisting of either two β subunits (activins: βAβA, βBβB and βAβB) or one β and one unique α subunit (inhibin: αβ), all belonging to the TGF-β superfamily [81]. All three β subunits (βA: *inhbaa*, *inhbab*; and βB: *inhbb*) and *inha* were exclusively expressed in zebrafish follicle cells [52,53,67,82]. A recent single cell transcriptome analysis showed that *inha* and *inhbaa* were co-expressed primarily in the granulosa cells of follicles from 40-dpf fish [83], suggesting that *inhbaa* is likely the major form of β subunits responsible for inhibin production in addition to its self-dimerization to form activin Aa [67]. Disruption of *inha* would cause a complete loss of inhibin, which would not only divert Inhbaa towards producing more activin Aa (βAaβAa) but also reduce inhibition of activin action by inhibin, resulting in an overall increase in activin activity. We have proposed previously that activins from the somatic follicle cells may represent a major intrafollicular paracrine pathway that mediates the signaling from the follicle cells to oocyte [8]. It is conceivable that the resumption of vitellogenic growth in the double mutant (*bmp15-/-;inha-/-*) might involve enhanced activin activity especially from *inhbaa*. This idea is further supported by the evidence that the resumed vitellogenic growth in *bmp15-/-; inha-/-* was reduced by the loss of *inhbaa* in the triple mutant (*bmp15-/-;inha-/-;inhbaa-/-*), suggesting a critical mediating role for *inhbaa* in Bmp15 regulation of follicle growth. This is supported by the observation that the expression of *inhbaa* was significantly reduced in *bmp15-/-* PV follicles.

The role of activin in rescuing the *bmp15-/-* phenotype by *inha-/-* is further supported by the observation that, despite being the major form of the β subunit that forms inhibin in zebrafish [67], the loss of *inhbaa* was unable to rescue *bmp15-/-* deficiency in the double mutant *bmp15-/-;inhbaa-/-*, unlike *bmp15-/-;inha-/-*. This finding strongly supports our hypothesis that the increased production and activity of activin from *inhbaa* in *bmp15-/-;inha-/-* is likely one of the main underlying factors responsible for the partial rescue of *bmp15-/-* phenotypes by *inha-/-*. In comparison to *bmp15-/-;inha-/-* mutant, which lack inhibin but have elevated activin derived from Inhbaa, the double mutant *bmp15-/-;inhbaa-/-* would have lost both inhibin and activin Aa. Our previous studies have shown that the oocyte is a direct target for activin as evidenced by its expression of all activin receptors and Smad proteins as well as phosphorylation of Smad2 in response to activin treatment [22,53,82]. In addition to oocytes, activin might also participate in the regulation of *cyp19a1a* expression in the follicle cells as suggested by the evidence that the loss of *inhbaa* in the triple mutant (*bmp15-/-;inha-/-; inhbaa-/-*) abolished the increased expression of *cyp19a1a* in the double mutant (*bmp15-/-; inha-/-*) together with reduced E2 and Vtg levels in the serum. Whether activin works alone or together with other factors such as pituitary FSH and Bmp15 to regulate *cyp19a1a* remains unknown, and it will be an interesting issue to study in the future. In mammals including humans, activin has been widely reported to increase aromatase activity and estrogen production in cultured granulosa cells alone or together with FSH [84–88]. Similar to activin, the oocyte-derived BMP15 and GDF9 also significantly potentiated the actions of FSH in stimulating aromatase expression and estrogen biosynthesis in human granulosa cells [80] (Fig 11C).

One interesting observation was that although the lack of *inhbaa* in the triple mutant (*bmp15-/-;inha-/-;inhbaa-/-*) prevented yolk granule formation as seen in the double mutant (*bmp15-/-;inha-/-*), the oocytes could continue to grow to the size close to the EV stage in the absence of yolk granules and without formation of additional cortical alveoli. This has led us to hypothesize that the simultaneous mutation of *inha* and *inhbaa* resulted in the loss of both inhibin and activin Aa. Since inhibin antagonizes both activins and BMPs [89], the loss of

inhibin may also enhance the signaling by other forms of activins and BMPs despite the loss of activin Aa, which may promote oocyte growth. This is supported by our transcriptome data that in addition to *inhbaa*, both BMP type II receptors (*bmpr2a* and *bmpr2b*) and one activin receptor-like kinase 1 (*acvrl1*) were increased in both *inha-/-* and *bmp15-/-;inha-/-* follicles. This hypothesis will be tested in future studies. Our observation also suggests that although oocyte growth in fish species is associated with the accumulation of cortical alveoli and yolk granules, it can occur independent of the latter.

## Evidence for Bmp15 regulation of vitellogenin uptake–roles of vitellogenin receptors

The lack of vitellogenic growth in *bmp15* mutant females may be due to deficiencies in either Vtg biosynthesis in the liver as discussed above or Vtg uptake by the growing oocytes in the ovary or both. Compared to our knowledge about Vtg expression and regulation, our understanding of Vtg uptake remains rather limited in fish [76]. It is well known that Vtg proteins in oviparous vertebrates including fish are taken up by the growing oocytes from the blood stream via endocytosis mediated by Vtg receptors, processed and deposited in yolk granules [90]. Despite some studies on potential Vtg receptors in fish [91–94], the molecular nature of these receptors remains to be characterized. The first putative Vtg receptor in fish was cloned in the rainbow trout by homology-based PCR cloning and demonstrated to belong to low-density lipoprotein receptor (LDLR) superfamily [95], which also includes a large number of LDLR-related proteins (LRP). These proteins are primarily involved in endocytosis of various ligands, in particular the lipoproteins [96]. In striped bass and white perch, Lrp13 was cloned and demonstrated to be a potential Vtg receptor responsible for endocytosis-mediated Vtg uptake [97], and this was also confirmed later in the cutthroat trout [98].

In the zebrafish, eight Vtg genes (*vtg1-8*) have been identified in the genome [99]. Although Vtg receptors have not been characterized in zebrafish, they are most likely members of the LRP family according to the studies in other teleosts [97,98]. Interestingly, our transcriptome data revealed that both endocytosis and receptor-mediated endocytosis pathways were enhanced in *inha-/-* and *bmp15-/-;inha-/-* follicles, and the genes showing significant increase in expression in the receptor-mediated endocytosis pathway were all members of the LRP family including *lrp1ab*, *lrp2a*, *lrp5* and *lrp6*. These genes displayed distinct expression profiles during folliculogenesis, which are closely associated with the vitellogenic growth of the oocyte. The exact roles of these genes in vitellogenesis or Vtg uptake remain to be elucidated. Their significant increase in expression in the *bmp15-/-; inha-/-* follicles has led us to hypothesize that in addition to increasing *cyp19a1a* and *inhbaa* expression, resulting in increased Vtg biosynthesis and oocyte growth, the loss of *inha* may also enhance Vtg update at the follicle level, contributing to the rescue of failed yolk accumulation in the *bmp15-/-* mutant. Since inhibin is an antagonist of activin, which acts directly on the oocyte via Smad2/3 [22], we speculate that the increased activity of activin in *inha-/-* could be one of the factors responsible for the up-regulation of Vtg receptors in the oocyte (Fig 11C). This idea is supported by the evidence that the loss of *inhbaa* in the triple mutant (*bmp15-/-;inha-/-;inhbaa-/-*) abolished *lrp* expression (*lrp1ab* and *lrp2a*) and prevented yolk formation.

## Differential roles of *bmp15* and *gdf9* in controlling folliculogenesis

Gdf9 and Bmp15 are closely related molecules expressed specifically in the oocyte. Their interaction in biosynthesis and function has been studied in both mammals and fish [41,100]. In

addition to forming homodimers (GDF9 and BMP15), they can also form heterodimers (GDF9:BMP15), which exhibit much more potent bioactivities than the homodimers [100]. Our recent study on *gdf9* [22] and the present study on *bmp15* showed that although these two factors are both expressed in the oocyte, they play distinct roles in early folliculogenesis by controlling different developmental stages in a sequential manner. The loss of *gdf9* resulted in a complete arrest of follicle development at PG stage without formation of cortical alveoli [22], whereas disruption of *bmp15* gene blocked follicles at PV stage with formation of cortical alveoli but not yolk granules (Fig 11C).

To further explore the relationship between Gdf9 and Bmp15, we created a double mutant of the two genes (*bmp15-/-;gdf9-/-*). The double mutant basically phenocopied *gdf9-/-*, similar to *Bmp15-/-;Gdf9-/-* in mice [32]. Interestingly, follicles in some double mutant females could break the blockade seen in *gdf9-/-* single mutant to enter early PV stage with formation of small cortical alveoli (PV-I); however, they could not develop further to late PV stages (PV-II and III) as seen in *bmp15-/-* single mutant. This is surprising and raises an interesting question about the interaction of Gdf9 and Bmp15 in follicle activation or PG-PV transition. Our hypothesis is that Gdf9 and Bmp15 may play antagonistic roles in controlling the formation of cortical alveoli, the marker for PG-PV transition, and they act in a sequential manner. At the onset of PG-PV transition, the oocyte may first release Gdf9, which stimulates the biogenesis of cortical alveoli in the oocyte. This is followed by release of Bmp15 at the onset of PV-EV transition, which stimulates vitellogenesis (vitellogenin synthesis and update) while suppressing the formation of cortical alveoli to create space for yolk granules (Fig 11D). Since both Gdf9 and Bmp15 are oocyte-derived factors that act on the surrounding follicle cells, their regulation of oogenesis (formation of cortical alveoli and yolk granules) is likely mediated by factors from the follicle cells in a paracrine manner. The double mutation of *gdf9* and *bmp15* would mean simultaneous loss of both stimulation and inhibition of cortical alveoli, resulting in advancement of some follicles into early PV stage (PV-I), a phenotype between the characteristics of *gdf9-/-* (complete arrest at PG stage) and *bmp15-/-* (fully developed PV stage). This hypothesis will need to be tested and verified by experiments in the future. The idea that Gdf9 and Bmp15 may act in an antagonistic manner has also been proposed in a recent study in Japanese flounder. Overexpression of Gdf9 and Bmp15 in a flounder ovarian cell line could up-regulate and down-regulate steroidogenic enzymes via Smad2/3 and Smad1/5/8 pathways, respectively [41]. The factors from the follicle cells that mediate Gdf9 and Bmp15 actions remain unknown; however, our data suggest activin as one potential candidate. Recombinant zebrafish Gdf9 could stimulate expression of activin (*inhbaa* and *inhbb*) in cultured follicle cells probably via Smad2/3 pathway [16] and the loss of *gdf9* gene resulted in a significant decrease in *inhbaa* expression [22]. Similarly, the present study showed that the expression of *inhbaa* also decreased significantly in the PV follicles of *bmp15* mutant (Fig 11D).

In summary, using genetic analysis involving multiple mutants including *bmp15*, *gdf9*, *inha* and *inhbaa* as well as physiological and pharmacological approaches, the present study provided critical insights into the mechanism underlying Bmp15 actions, its interaction with the activin-inhibin system, and functional relation with Gdf9 in controlling folliculogenesis. To our knowledge, this represents the most comprehensive genetic study in vertebrates on BMP15, an oocyte-derived growth factor that plays important roles in orchestrating follicle development. Our data strongly support an important role for Bmp15 in controlling vitellogenic growth of follicles in zebrafish and provide critical evidence for its regulation of vitellogenin production via estrogens as well as vitellogenin uptake in the ovary by the growing oocyte via vitellogenin receptors. All these actions involve participation of TGF-β family members, particularly activin and inhibin.

## Materials and methods

### Ethics statement

The animals were handled according to the Animal Protection Act enacted by the Legislative Council of Macao Special Administrative Region under Article 71(1) of the Basic Law and all experimental protocols were approved by the Research Ethics Panel of the University of Macau (AEC-13-002).

### Zebrafish maintenance

The AB strain zebrafish (*Danio rerio*) was used to generate mutant lines in the present study. The fish were kept at 28 ± 1 ˚C with a photoperiod of 14-h light and 10-h dark in the ZebTEC Multilinking Rack zebrafish system (Tecniplast, Buguggiate, Italy). The adult fish were fed twice daily with artemia and Otohime fish diet (Marubeni Nisshin Feed, Tokyo, Japan), which was delivered by the Tritone automactic feeding system (Tecniplast). The larval fish were fed with paramecia (5–10 dpf) and artemia (10–20 dpf) before transfer to the main system.

### Generation of *bmp15* null mutant line

The CRISPR/Cas9 method was used to generate *bmp15* mutant line in zebrafish according to the protocols reported previously [101–103]. Briefly, a single-guide RNA (sgRNA) (S4 Table) targeting the exon I of *bmp15* gene was designed by using the online tool (http://zifit.partners.org/ZiFiT/Disclaimer.aspx). The Cas9 mRNA and sgRNA were produced by *in vitro* transcription from pCS2-nCas9n (Addgene Plasmid #47929) and DraI-digested pDR274 (Addgene Plasmid #42250) using the mMACHINE T7 kit and mMACHINE SP6 kit according to the manufacturer's instruction (Invitrogen, Carlsbad, CA). Both Cas9 mRNA (300 ng/μL) and sgRNA (75 ng/μL) were co-microinjected (4.6 nL) into zebrafish embryos at one- or two-cell-stage using the Drummond Nanoject injector (Drummond Scientific, Broomall, PA).

### Genotyping by high-resolution melting analysis (HRMA) and heteroduplex mobility assay (HMA)

The genomic DNA was extracted by the NaOH method from a single embryo or piece of caudal fin as described preciously [101,104]. Briefly, 40 μL NaOH with the concentration of 50 nmol/μL was added in the tube containing one embryo or a caudal fin piece and incubated at 95˚C for 10 min. Then 4 μL Tris-HCl (pH 8.0) was added to neutralize the reaction. HRMA is a powerful tool to screen indel mutations or single nucleotide polymorphism in the samples and it was performed on the genomic DNA with specific primers listed in S4 Table using the CFX96 Real-Time PCR Detection System and the Precision Melt Analysis software (Bio-Rad Laboratories, Hercules, CA). HMA was performed with PAGE to verify the genotyping results of HRMA. Briefly, the HRMA product (5 μL) was subjected to electrophoresis in 20% polyacrylamide gels for 5 h at 100 V. Then the gel was stained with GelRed (Biotium, Hayward, CA) and visualized on the ChemiDoc imaging system (Bio-Rad). Because HMA can detect small changes in sequence, this method is more sensitive than HRMA but with low throughput. The heterozygotes were detected as two bands and homozygotes showed only one band.

### Sampling and histological examination

The fish were sampled for phenotype analysis at different time points of development and the body weight (BW) and standard body length (BL) were recorded. Histological analysis was performed on paraffin sections using hematoxylin and eosin (H&E) staining. Briefly, the

fish were anaesthetized with MS-222 (Sigma-Aldrich, Louis, MO) before measuring BL and BW by ruler and analytical balance respectively. Then the fish were photographed for gross morphology with a digital camera (EOS700D, Canon, Tokyo, Japan). The cloaca and pectoral fin were observed under the Nikon SMZ18 dissecting microscope and photographed with the Digit Sight DS-Fi2 digital camera (Nikon, Tokyo, Japan) for genital papilla (GP) in females and breeding tubercles (BT) in males. After these examinations, the fish were fixed in Bouin's solution for at least 24 h. Dehydration and paraffin imbedding were then performed on the ASP6025S Automatic Vacuum Tissue Processor (Leica, Wetzlar, Germany). The samples were sectioned using the Leica microtome (Leica) at 5 μm thickness. After deparaffinization, hydration and staining, the sections were examined on the Nikon ECLIPSE Ni-U microscope and micrographs were taken with the Digit Sight DS-Fi2 digital camera (Nikon).

## Follicle staging and quantification

The follicles on sections were staged according to both size (diameter with visible germinal vesicle) and morphological features (cortical alveoli and yolk granules) as previously reported [55] and they were divided into the following six stages: primary growth (PG, stage I; <150 μm), previtellogenic (PV, stage II; ~250 μm), early vitellogenic (EV, early stage III; ~350 μm), mid-vitellogenic (MV, mid-stage III; ~450 μm), late-vitellogenic (LV, late stage III; ~550 μm) and full-grown (FG; >650 μm). To quantify follicle composition in the ovary, we performed serial longitudinal sectioning of the whole fish at a thickness of 5 μm. We measured the follicle diameters on three sections that were spaced at least 60 μm apart using the NIS-Elements BR software (Nikon). Among the three sections, the middle one represented the largest in the series. To ensure accuracy of diameter measurement for follicle staging, we only measured the follicles with visible nuclei (germinal vesicles) on the section.

## Sex identification

The phenotypic sex of zebrafish was identified according to dimorphic morphological features including body shape, fin color, BT on the pectoral fin and GP at the cloaca. Normally, male fish showed a slim body shape with brownish color, clear BTs on the pectoral fin and an invisible GP, while female fish had a round body shape with silverish color, no BT, and a prominent GP. The area of BTs on the third pectoral fin ray was quantified by the ImageJ software. The sex of each fish was confirmed by histological sectioning and microscopic observation. The fish with well-formed ovaries were identified as females, while the fish with severely degenerating ovaries with or without testicular tissues were taken as intersexual type and the fish with well-formed testis were identified as males.

## Fertility assessment

The number of ovulated eggs and the percentage of survived embryos (fertilization rate) are two indicators used for assessment of fish fertility when they were mated with wild type (WT) partners through natural spawning. The fertility tests were performed on 4-mpf fish at 5-day interval, and each test involved five females of each genotype (+/+ or −/−) and five WT males in a breeding tank. The number of ovulated eggs was counted and averaged per female within three hours after spawning and the number of surviving embryos (transparent) was counted after 24 h. Individuals that failed to produce fertilized embryos after at least 10 trials were considered infertile.

## Measurement of serum E2 and Vtg levels

To determine serum concentrations of E2 and Vtg, the blood was sampled from the heart directly with a glass capillary according to our reported protocol [105,106]. The blood samples were collected without treating the capillaries with heparin sodium, and they were kept for half an hour at room temperature to separate the serum. The supernatant (serum) was carefully transferred to a new microtube after centrifugation (5000 rpm, 20 min, 4˚C). The levels of E2 and Vtg in the serum from each fish were measured using the ELISA kits for E2 (Neogen, Lansing, MI) and zebrafish Vtg (Cayman Chemical, Ann Arbor, MI), respectively, according to the manufacturer's instructions.

## E2 and fadrozole treatment

17β-estradiol (E2, CAS: 50-28-2, $\geq$ 99%) and fadrozole (CAS: 102676-31-3, $\geq$ 98%), a nonsteroidal aromatase inhibitor, were purchased from Sigma-Aldrich (St. Louis, Missouri). The *bmp15*-null female fish were treated with E2 for 20 days (from 80 to 100 dpf) continuously. We used two treatment methods: water-borne exposure and oral administration by feeding. For water-borne exposure, the *bmp15* mutant fish were placed in a clean tank with 10 L water. E2 stock solution or the vehicle ethanol was added to the water to the final concentrations of 0 and 10 nM. The water was renewed daily during the exposure period to maintain relatively constant concentrations. For oral administration, the Otohime fish feed was mixed with E2 stock solution and dried overnight in an oven at 60˚C. The fish were fed twice a day with the dried feed containing E2 at different concentrations (0, 2, 20 and 200 μg/g) and supplemented with artemia. The total amount of feed administered was 10% (W/W) of fish body weight per day (5% per meal). For the double mutant females (*bmp15-/-;inha-/-*), they were treated similarly from 80 to 100 dpf with dried powder feed containing fadrozole (0 and 200 μg/g).

## Transcriptome analysis

To investigate the mechanisms of Bmp15 actions, we collected PG follicles from 12 samples at 4 mpf as follows: three control fish (*bmp15+/+;inha+/+*), three *bmp15* mutant fish (*bmp15-/-*), three *inha* mutant fish (*inha-/-*) and three double mutant fish (*bmp15-/-;inha-/-*). All female fish were genotyped by HRMA, and their ovaries were carefully removed and placed in a microtube containing 1 mL L15 medium (Sigma-Aldrich). We used a BD Microlance 26G needle (BD, San Diego, CA) to remove fat tissues and ligaments surrounding the ovaries. The ovarian fragments were gently pipetted several times using a transfer pipette (JET Biofil, Guangzhou, China) to separate the follicles. The PG follicles were then isolated by passing the isolated follicles through a 150 μm filter (SPL Lifesciences, Waunakee, WI). The isolated PG follicles (< 150 μm) were collected by centrifugation at 3500 rpm (5 min) for transcriptome (RNA-seq) analysis. The RNA extraction, cDNA library construction, quality control, RNA sequencing and preliminary analysis of transcriptome data were carried out using Novaseq 6000 sequencing system (Novogene Bioinformatics Technology, Tianjin, China). Genes with an adjusted P-value $\leq$ 0.05 were defined as differentially expressed genes (DEGs) with statistical significance. DEGs were then subject to Gene Ontology (GO) enrichment analysis and Kyoto Encyclopedia of Genes and Genomes (KEGG) pathway enrichment analysis using Cluster Profiler R package. The RNA-seq data are available at NCBI under the BioProject No. PRJNA849009.

## RNA extraction and quantitative real-time qPCR

Total RNA was isolated from various tissues using TRIzol (Invitrogen), and quantified using NanoDrop 2000 (Thermo Scientific, Waltham, MA) based on the absorbance at 260 nm. To

quantify the levels of gonadotropin expression (*fshb* and *lhb*) in each individual pituitary, the gland from each fish was removed immediately following decapitation for RNA extraction and real-time qPCR analysis according to our previous report [4]. Reverse transcription reaction was performed on the same amount of total RNA using M-MLV reverse transcriptase (Invitrogen). Real-time qPCR was performed on the CFX96 Real-Time PCR Detection System using the SsoFast EvaGreen Supermix (Bio-Rad). Each sample was assayed in duplicate for accuracy, and the primers used are listed in S4 Table. The expression of target genes in each sample was normalized to that of the housekeeping gene *ef1a*, and expressed as the ratio to the control group.

## Double-colored fluorescent *in situ* hybridization (FISH)

To detect the expression of gonadotropins (FSH and LH) in the mutant, we performed double-colored fluorescent FISH analysis. The cDNAs of *fshb* and *lhb* were amplified by specific primers as described before [4]. The sense and anti-sense probes were prepared from pBluescript II KS containing cDNAs of *fshb* and *lhb*. Both probes were labeled with fluorescein or digoxigenin (DIG) using RNA labeling kit (Roche Applied Science, Mannheim, Germany). The heads of fish at 4 mpf were fixed in 4% paraformaldehyde (PFA) for at least 48 h, embedded in paraffin and sectioned at 5 μm thickness. The sections were deparaffinized and rehydrated before digestion with proteinase K (4 μg/ml; Roche) for 15 min at 37˚C. The sections were then hybridized with fluorescein and DIG-labeled RNA probes overnight at 60˚C. On the following day, the sections were first washed with 5× saline-sodium citrate for 5 min, 2× SSC for 20 min, and 0.2× SSC for 20 min (SSC; 0.15 M NaCl and 0.015 M sodium citrate). The hybridization signal was detected by horseradish peroxidase (HRP)-conjugated anti-fluorescein antibody (Roche) with TSA-fluorescein. The sections were incubated in 1% $H_2O_2$ for 60 min to deactivate the HRP from the first staining before detecting the second signal. Then the HRP-conjugated anti-DIG antibody (Roche) was added and the signal was detected by TSA-cy5/TMR system. Prolong Gold antifade reagent (Invitrogen) was used for mounting the sections, which were viewed on the Olympus FluoView1000 confocal microscope (Olympus, Tokyo, Japan) for image analysis.

## Statistical analysis

All values in this study were obtained from multiple independent experiments and/or biological repeats (n≥3). The data are presented as mean ± standard error of the mean (SEM), and the significance (*P<0.05; **P<0.01; ***P<0.001; n.s., not significant) was analyzed by Student's t-test or ANOVA followed by Tukey's multiple comparison. Chi-squared ($X^2$) test was used for sex ratio analysis. All analyses were performed with Prism 8 (GraphPad Prism, San Diego, CA).

## Supporting information

**S1 Fig. Spatiotemporal expression of *bmp15* and other genes in the follicle and during folliculogenesis.** (A) Spatial distribution of gene expression in the FG follicle. The housekeeping gene *ef1a* was expressed in both oocyte and follicle layer, whereas the marker genes *lhcgr* and *gdf9* were expressed in the follicle layer and denuded oocyte respectively, indicating clean separation of the two compartments. The expression of *bmp15* was detected exclusively in the denuded oocyte, whereas *inha*, *inhbaa* and two BMP type II receptors (*bmpr2a* and *bmpr2b*) were detected only in the follicle layers. In addition, *lrp1ab*, *lrp5* and *lrp6* were all expressed in both follicle layer and oocyte whereas *lrp2a* was exclusively expressed in the follicle layer. (B) Temporal expression profiles of *bmp15* and other genes during folliculogenesis. The

expression patterns of *fshr* and *lhcgr* were used as the internal reference for appropriate staging of the follicles. The relative mRNA levels were determined by real-time qPCR, normalized to *ef1a*, and expressed as fold change compared with the levels at the PG stage. The values are the mean ± SEM (n = 3 fish) from a representative experiment and analyzed by ANOVA followed by the Tukey HSD for multiple comparisons. Different letters indicate statistical significance (P < 0.05). PG, primary growth; PV, previtellogenic; EV, early vitellogenic; MV, mid-vitellogenic; LV, late vitellogenic; FG, full-grown.
(TIF)

**S2 Fig. Mutagenesis of *bmp15* and mutant characterization.** (A) Schematic illustration of the genomic structure of zebrafish *bmp15* gene. The black and white boxes indicate the coding and non-coding regions of the exon, respectively. The underlined sequence indicates CRISPR/Cas9 target site, and the dashed line indicates the deleted sequence (-5 bp) of zebrafish *bmp15*. The primers for mutation screening (*bmp15_5284_F/bmp15_5285_R*) are shown below. (B) Schematic representation of Bmp15 amino acid (aa) sequence. The 5-bp deletion introduced a premature termination codon, resulting in the synthesis of a truncated protein of 55 aa. (C) Confirmation of mutation at the mRNA level in the ovary. RT-PCR was performed on total RNA extracted from WT, heterozygous and homozygous mutant ovaries with a specific primer (*bmp15_6481_F/bmp15_5285_R*) overlapping with the deleted sequence. No signal could be detected in the mutant ovary. (D) Genotyping with HRMA. Melt curves for WT, heterozygotes and homozygous mutant are marked in black, green, and red respectively. (E) Genotyping with HMA. The heterozygotes showed two additional bands and the homozygous mutant showed a smaller band than WT.
(TIF)

**S3 Fig. Blocked folliculogenesis in *bmp15*-deficient females.** (A) Histology analysis of the control (*bmp15+/-*) and mutant (*bmp15-/-*) fish at 120 dpf. Follicle development was arrested at PV stage in *bmp15-/-* mutant. (B) Size distribution of follicles in the control (*bmp15+/-*) and mutant (*bmp15-/-*) fish at 120 dpf (n = 3). The mutant follicles could grow to the size of PV stage only (~250 μm), while the control follicles could reach FG stage (> 650 μm). (C) Follicle composition analysis at 120 dpf. Compared with the control, the *bmp15-/-* ovary contained significantly more PG (stage I) follicles but no vitellogenic follicles (EV–FG, stage III). The values are expressed as mean ± SEM (n = 6) and analyzed by ANOVA followed by the Tukey HSD for multiple comparisons (*** P < 0.001).
(TIF)

**S4 Fig. Histogram of the top enriched GO terms for up-regulated and down-regulated DEGs.** PG follicles from the wild type (WT) control (*bmp15+/+;inha+/+*) and mutants (*bmp15-/-*, *inha-/-* and *bmp15-/-;inha-/-*) were used for RNA-seq analysis. GO terms include three parts: biological process (BP), cellular component (CC) and molecular function (MF).
(TIF)

**S5 Fig. Expression of 20 selected DEGs from the TGF-β signaling, endocytosis and receptor-mediated endocytosis pathways.** The DEGs were identified by RNA-seq analysis and confirmed by RT-qPCR.
(TIF)

**S6 Fig. Histological analysis of the control (*bmp15+/-;inha+/-;inhbaa+/-*) and double mutants (*bmp15-/-;inhbaa-/-* and *inha-/-;inhbaa-/-*) at 120 dpf.** The follicle development was arrested at PV stage in *bmp15-/-;inhbaa-/-* double mutant while the follicles showed normal vitellogenic growth in *inha-/-;inhbaa-/-* double mutant. PG, primary growth; PV, pre-

vitellogenic; FG, full-grown.
(TIF)

**S7 Fig. Testis development and spermatogenesis in *bmp15*, *inha*, and *inhbaa* mutant males.** (A) Morphology, gross anatomy, and histological analysis of sexually mature males at 180 dpf. The boxed areas are shown at higher magnification below. (B) Area of the spermatozoa-filled luminal spaces in the testis. There was no significant difference among these mutants compared with the control at 180 dpf. The values are expressed as mean ± SEM (n = 5) and analyzed by ANOVA followed by the Tukey HSD for multiple comparisons. (C) Offspring of control (*bmp15*+/–) and mutant (*bmp15*–/–) males with WT females (*bmp15*+/+) at 3 hpf and 72 hpf.
(TIF)

**S1 Table. RNA-seq data on *bmp15*-/- vs. control (*bmp15*+/+;*inha*+/+).**
(XLSX)

**S2 Table. RNA-seq data on *inha*-/- vs. control (*bmp15*+/+;*inha*+/+).**
(XLSX)

**S3 Table. RNA-seq data on *bmp15*-/-;*inha*-/- vs. control (*bmp15*+/+;*inha*+/+).**
(XLSX)

**S4 Table. Primer used for CRISPR, HRMA and RT-PCR.**
(DOCX)

## Acknowledgments

We thank Ms. Phoenix Un Ian LEI for the maintenance and management of the zebrafish facility and the Histology Core of the Faculty of Health Sciences for technical support.

## Author Contributions

**Conceptualization:** Yue Zhai, Wei Ge.

**Data curation:** Yue Zhai, Xin Zhang, Wei Ge.

**Formal analysis:** Yue Zhai, Wei Ge.

**Funding acquisition:** Wei Ge.

**Investigation:** Yue Zhai, Xin Zhang, Cheng Zhao, Ruijing Geng, Kun Wu, Mingzhe Yuan, Nana Ai, Wei Ge.

**Methodology:** Yue Zhai, Xin Zhang, Cheng Zhao, Ruijing Geng, Kun Wu, Mingzhe Yuan, Nana Ai.

**Project administration:** Nana Ai, Wei Ge.

**Resources:** Wei Ge.

**Validation:** Yue Zhai, Xin Zhang, Wei Ge.

**Visualization:** Yue Zhai, Xin Zhang, Wei Ge.

**Writing – original draft:** Yue Zhai, Wei Ge.

**Writing – review & editing:** Yue Zhai, Wei Ge.

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
