## [Decision Letter · Decision Letter 0]

2 Apr 2023

Dear Dr Ge,

Thank you very much for submitting your Research Article entitled 'Rescue of BMP15 ( bmp15 ) Deficiency by Mutation of Inhibin a (inha) Provides Novel Clues to How Bmp15 Controls Zebrafish Folliculogenesis' to PLOS Genetics.

The manuscript was fully evaluated at the editorial level and by independent peer reviewers. The reviewers appreciated the attention to an important problem, but raised some concerns about the current manuscript. Many good comments are provided by the reviewers to improve the data presented in the figures, validate by RT qPCR some of the RNAseq results, as well as clarifications to the text.  Based on the reviews, we will not be able to accept this version of the manuscript, but we would be willing to review a much-revised version.

If you decide to revise the manuscript for further consideration at PLOS Genetics, please aim to resubmit within the next 90 days, unless it will take extra time to address the concerns of the reviewers, in which case we would appreciate an expected resubmission date by email to plosgenetics@plos.org.

Please do not hesitate to contact us if you have any concerns or questions.

Yours sincerely,

Mary C. Mullins

Academic Editor

PLOS Genetics

Gregory Barsh

Editor-in-Chief

PLOS Genetics

Reviewer's Responses to Questions

**Comments to the Authors:**

Reviewer #1: In this study authors investigate the mechanism by which BMP15 regulates zebrafish follicle development. They generated single (bmp15-/-), double (bmp15-/-;inha-/-) and triple (bmp15-/-;inha-/-;inhbaa-/-) mutants to study the interaction of Bmp15 with avidin/inhibin. The evidence provided from genetic and transcriptomic approaches is solid. However, the new knowledge is embedded in many figures of known information, preventing easy understanding of the paper. The title of the paper also does not deliver clear message about their new findings. Authors should differentiate their new findings from those already known from the literature. For example, it will be nice to simplify their summary figure (Fig. 13) and differentiate their new information from known knowledge. Since many of these mutants have been reported, some known data can be moved to the supplement.

Major point:

1. Authors hope to identify pathways that participate in the rescue of bmp15 mutants by inha mutation using RNAseq in Fig. 6, but the results are not clear. The pathways they identify, eg. TGF-beta and receptor-mediated endocytosis, do not seem to be reduced in bmp15 mutants. More detailed statistical comparison will be necessary.

Furthermore, these data will need to be validated by real-time qPCR. Further follow-up experiments or at least some explanation will be required to dissect the relationship of bmp15, inha, and ovary development.

Please use the same color codes for all samples throughout Fig. 6A-C to facilitate reading. In Fig.6C, the Y-axis of the normalized read count in the TGF-beta signaling pathway should be normalized and presented as a log2 scale.

2. Figure 9B: Authors show that cyp19a1a mRNA is increased by E2 treatment in the PG of bmp15 mutants. E2 is produced in PV. It would be more appropriate to examine PV.

3. The presentation of Figure 11A-D is confusing. One cannot tell which genes are up- or down-regulated from the colors. The scale bars representing mRNA levels may need to be normalized against those of the control fish and the fold change presented as a log2 scale. Which two groups of samples are compared to obtain the asterisk?

Minor:

1. Line 295, please add back the parenthesis for “Fig6A”.

2. The image of 50-dpf bmp15-/- mutant in Fig.2A is not representative. Fig. 2F shows that most gonads (5/7) already have PV.

3. It will be better to present the X- and Y-axes of Figure 2C and 2D as the percentage of gonads with certain follicular stage versus age, then the temporal delay is more evident. How was the ratio of delayed puberty onset calculated? The method of calculation should be briefly described in the legend and in more detail in the M&M section. The scale bar should also be labeled in Fig. 2B.

4. Figure 4B: for easier understanding, the arrow indicating GP and BTs should be in different colors.

5. Please describe the morphology of Fig. 10B in text L399.

6. The abbreviations of EPN and LPN should be mentioned in Fig.13B legend.

7. Figure S1A: the meaning of black and white boxes should be described in figure legend.

Reviewer #2: The manuscript by Zhai et al, investigates the relationship between BMP signaling and inhibin in oogenesis. The authors utilize elegant genetic experiments to demonstrate that BMP15 and activin/inhibin signaling control key events in oogenesis. Furthermore, they demonstrate that E2 production is a major output of these signaling pathways necessary for oogenesis as well as vitellogenesis production and uptake, which are likely downstream to E2. In addition, the authors compare the relative roles of GDF9 and BMP15 and, through double mutant analysis, show that they may have some antagonistic roles. The authors Overall the experiments are thorough and support the conclusions made by the authors. This is a robust study and important body of work that enhances knowledge on signaling networks that are essential for oogenesis. The manuscript is clearly written and will be of broad interest to reproductive biologists and particularly fish reproductive biologists.

Specific comments:

Line 50: typo – “update” should be “uptake”

Lines 255 – 263: It would be very interesting to follow the progression of sex reversal in the same fish using external landmarks (GP and BTs).

In the RT-PCR experiments shown in figure 1A , what stage/age were oocytes/ovaries that RNA was isolated from?

Line 78: there were not significantly different changes in gene expression between bmp15 mutant and bmp15; inha double mutants. Treatment with E2 and fadrozole also did not show a statistically significant change in vtg3 expression

Lines 88 – 91: Was fertility tested in the bmp15 mutant males?

Figure S1: Panel C is missing

Reviewer #3: Review of Rescue of BMP15 ( bmp15 ) Deficiency by Mutation of Inhibin a (inha) Provides Novel Clues to How Bmp15 Controls Zebrafish Folliculogenesis

General comments

To understand the roles of the oocyte signaling molecule Bmp15, Zhai et al. made a new allele of the zebrafish bmp15 gene, studied it’s phenotype, and then advanced understanding by making double mutants with inhibin- and activin-encoding genes and the related gdf9 gene. In addition, they treated zebrafish with estrogen and an aromatase inhibitor to knockdown estrogen levels and examined the levels of vitellogenin yolk proteins in the blood. Results show that Bmp15 exerts its effect to allow vitellogenesis by acting through the activin/inhibin system mediated by estrogen. The results provide the fullest understanding of Bmp15 action in the development of ovarian follicles for any vertebrate.

One difficulty with interpretation is that, to be able to interpret the triple mutant result, we need to know the phenotype of the bmp15;inhbaa double mutant and the inha; inhbaa double mutant. These animals would have come out of the crosses performed to obtain the triple mutant. Otherwise, the experiments are extensive, well described and documented, and provide a nice advance.

The data tables for the RNA-seq results should be included in the supplementary data because the text gives the impression that the up- and down-regulated genes highlighted were selected to further the authors’ interpretations. The text gave no data on how up- or down-regulated those mentioned genes were with respect to other genes some other researchers might have decided to prioritize.

Specific comments

110 oocyte maturation whereas treatment with recombinant human BMP15 suppressed

human chorionic gonadotropin (hCG)

Does this imply that zebrafish has a human protein, or does it refer to some undescribed transgenic zebrafish, or to the zebrafish ortholog?

133 suggest that both Gdf9 and Bmp15 are [both] critical

136 activation or puberty onset (PG-PV transition).

Define ‘puberty’ for a fish.

151 Both gdf9 and bmp15 were detected exclusively in the denuded oocytes.

According to zebrafish nomenclature rules, this sentence says that the gdf9 gene was only in oocytes. But the zebrafish gene is not only in oocytes but also is in follicle cells, it’s just not expressed in follicle cells. The sentence should say: ‘Both gdf9 and bmp15 expression was detected exclusively in the denuded oocytes.’

See: https://zfin.atlassian.net/wiki/spaces/general/pages/1818394635/ZFIN+Zebrafish+Nomenclature+Conventions.

174 deletion was established that introduced an early terminator, [predicted to result] in synthesis …

Unless the protein was isolated and shown experimentally to be a prematurely short protein, then this is only a prediction.

175 The loss of bmp15 was also

The bmp15 gene was not lost. A few nucleotides were lost and its mRNA was depleted and its function was lost.

180 it is difficult to distinguish the homozygous mutant (bmp15–/–) from WT (bmp15+/+).

It’s not so difficult to detect the homozygous mutant: the bmp15 mutant ovary eventually becomes a testis. I think the authors mean here that the homozygous mutation is difficult to distinguish from the wild type gene. Or the mutant allele is difficult to distinguish from the wild-type allele. A mutant is an animal. A mutation is a change in DNA sequence.

203 crossed the thresholds (1.80 cm [or] 100 mg).

Just so it’s not a mathematical format for division.

206 control group (bmp15+/-)

Did the control group only have heterozygotes? Or did it have 2/3 heterozygotes and 1/3 homozygous wild types?

207 BW and BL; however, about 40% mutants (bmp15-/-)

Give actual numbers of fish as well as the %.

209 In addition, the mutant fish that [had] initiated puberty onset

212 The use of the ‘puberty onset’ concept doesn’t quite feel right to me. I’m not for sure set against it, but I don’t see what’s gained by it and think that the follicle growth dynamics vs. time or vs. weight and length is what’s important.

Fig. 2. C and D are a bit difficult to compare. In some ways, it would be nice to have the two genotypes displayed on the same graph. That has the advantage of first, making the comparison easier to see, even though the dotted lines do help, and second, of visualizing better whether the weight to length ratios of the two genotypes are the same. It has the disadvantage of a confusing mix of colors with so many points even if the two genotypes were shown in different shaped symbols. The other variable is time, and it would be nice to see the length and/or weight of both vs time to see if the mutation affects growth rate generally or if it is specific for the follicles.

216 histological examination of bmp15 mutants at sexual maturation stage (120 dpf).

Should the text also give weight and length for this stage?

217 control fish (bmp15+/+ and bmp15+/-) showed

Here, both phenotypically wild type genotypes are listed, but in other places, only heterozygotes. Were the controls really different in these instances? Or does the text not accurately describe control genotypes?

217 bmp15+/+ and bmp15+/-) showed normal folliculogenesis with full range

of follicles from PG to FG.

The Fig 3A homozygous wt image looks to have more advanced follicles than the heterozygous ovary. Does the mutation have a partially dominant effect?

230 During this period, the testis and spermatogenesis were normal in mutant males.

Some reviewers might insist on documentation and quantification for this claim, but I don’t think it’s necessary.

233 ovaries (8/21) and testes (10/21),

The writing says there were 8 of 21 ovaries and 10 of 21 testes. Did one of the presumably 11 fish examined have only one ovary so that there were 21 instead of 22 ovaries? Or does the sentence mean ovaries in 8 of 21 fish examined, etc.?

235 (11/19) with [some other] fish at transitional state

241 with a slim body shape, and they develop unique breeding tubercles (BT) on the

pectoral fins [60].

Add here that the BT depend on testosterone.

263 occurred asynchronously starting

Change to: occurred at different times in different individuals

275 cortical alveoli but not yolk granules in the oocytes, the double mutant bmp15-/-;inha-

/- showed normal vitellogenic growth beyond PV stage with large amount of yolk accumulated in the oocytes

this is an interesting and important observation

272 mutant bmp15-/-;inha-/- and examined its folliculogenesis

The text here must tell the reader what the phenotype is of the inha mutant compared to the bmp15 mutant before going into the phenotype of the double mutant.

284 Transcriptome analysis of bmp15-/-, inha-/- and bmp15-/-;inha-/- mutant follicles

In principle, the best experimental design would be to do transcriptomics also on the triple mutants and the inhbaa mutants and the inha;inhbaa double mutants, but for this work, I don’t think that’s necessary. The knowledge to be gained would be small compared to the expense and effort and should not hold up the publication of this work.

301 expression patterns suggested that during folliculogenesis, Bmp15 may act as a facilitator [of what?], while inhibin acts as a depressant [of what? Generalized transcription? Text needs to specify this.]

351 feeding at 200 μg/g and water-borne exposure at 10 nM suppressed ovarian

Provide here a brief interpretation phrase. Were the higher doses simply generally toxic for some reason?

Also, it is important to know the effect of the estrogen treatments on the genital papillae of females and sex tubercles of males. Were the higher doses of estrogen inhibitory to only the ovary or to secondary sex characteristics as well?

392 members. To test this hypothesis, we created a triple mutant using an inhbaa mutant

Text needs to tell reader the phenotype of the inhbaa single mutant.

392 To be able to interpret the triple mutant result, we need to know the phenotype of the bmp15;inhbaa double mutant and the inha; inhbaa double mutant. These animals would have come out of the crosses performed to obtain the triple mutant.

Fig. 10. I like the way that the red font makes it easier to tell which genes are homozygous for the mutation. A simple way to make it easier for the reader.

412 the femaleness was maintained, and sex change did not occur in both double and triple mutants

That’s an interesting and important observation. A single sentence here interpreting that finding would be helpful.

443 Since Vtg production in the liver is tightly controlled by estrogens from the ovary,

Document for zebrafish by citation.

Fig. 11. Scale bars should have units in the figure.

452 At the pituitary level, lhb showed no response

The pituitary isn’t mentioned in the materials and methods, which should be fixed.

460 signal for fshb in the triple mutant (Fig. 11E).

It would be helpful to have a phrase here that interprets this result in terms of feedback signaling between gonad and pituitary.

482 Follicle composition analysis showed [many fewer] vitellogenic follicles in the ovary

485 and 55.0% respectively while the ratio dropped significantly to only 22.6% in the triple

mutant (Fig. 12D).

It would be helpful to have a brief phrase here to interpret this observation.

490 observed either in the double mutant of bmp15 and inha

It would be helpful to remind reader of the spermatogenesis in inha and inhbaa single mutants here and to say in a phrase the significance.

513 The phenotypes of the published bmp15 mutant from the draper lab are different from the phenotype reported here in that the other allele allowed females to develop

572 The BTs seemed to be a sensitive marker for the onset of sex reversal as they appeared even

before any testicular tissues were visible in the ovary

This is interesting because the BTs arise due to testosterone. I would conclude that the testosterone level must increase before morphologies change detectably.

581 Our recent study showed that the loss of gdf9 gene in zebrafish

When the text uses the italicized gene name nomenclature, like gdf9 here, it means the gene, so the word ‘gene’ is not required after the gene name, here after gdf9.

582 arrest of follicle development at the PG stage.

Give citation.

642 Roles of [the] activin-inhibin system

643 One of [the] interesting findings

Fig. 13A. Nice summary figure. It would be good to put gdf9 on it as well as the various bmp15 variants.

746 The loss of gdf9 resulted in a complete arrest of follicle development at PG stage without formation of cortical alveoli

Show in fig13a.

Fig. 13B Many other mutants besides bmp15 and gdf9 result in oocytes dying and the juvenile ovary becoming a testis, including cyp19a1a, fancl, fancd1, foxl2a, foxl2b, igf3, and others. The position in 13B where some of these disrupt development should be indicated, how do these differ from the sex reversal in bmp15?

763 (vitellogenin synthesis and update) while suppressING the formation of cortical alveoli

advancement of some follicles into early PV stage (PV-I), somewhere between gdf9-/-

770 and bmp15-/-.

It’s unclear what is between these two mutant genotypes.

841 in males [respectivley (this word isn’t necessary and is misspelled)]

848 Follicle staging and quantification

Tell how these stages correspond to the Selman stages (https://doi.org/10.1002/jmor.1052180209) used by most in the field to improve comparison with other work.

856 largest sections spaced at least 60 μm apart

Does that mean 3 sections that are each 12 sections (=60µm/5 µm sections) apart? Or the individual follicles measured on each of the 3 sections were all 60 µm away from any other follicle measured on that section?

861 The [phenotypic] sex of zebrafish was identified…

871 Fertility assessment

Need to tell the age of the fish tested.

877 per female within three hours after spawning and the number of [surviving] embryos was…

How was living vs. dead determined at 24h? Maybe the embryo looked ok at 24h, but was actually dead?

912 medium (Sigma-Aldrich). We immediately dispersed the ovaries and isolated PG stage

For the method to be repeatable, the text has to tell how the ovaries were dispersed.

920 Cluster Profiler R package. The RNA-seq data are available at NCBI under the

BioProject No. PRJNA849009.

The authors must have the xl spreadsheet available as a supplemental table with this publication.

**Have all data underlying the figures and results presented in the manuscript been provided?**

Reviewer #1: Yes

Reviewer #2: Yes

Reviewer #3: Yes

PLOS authors have the option to publish the peer review history of their article (what does this mean?). If published, this will include your full peer review and any attached files.

Reviewer #1: No

Reviewer #2: No

Reviewer #3: No

---

## [Decision Letter · Decision Letter 1]

7 Aug 2023

Dear Dr Ge,

Thank you very much for submitting your Research Article entitled 'Rescue of bmp15 deficiency in zebrafish by mutation of inha reveals mechanisms of BMP15 regulation of folliculogenesis' to PLOS Genetics.

The manuscript was fully evaluated at the editorial level and by independent peer reviewers. The reviewers appreciated the changes and additions to the revised manuscript, but have noted some minor points that require addressing prior to acceptancet.

We therefore ask you to modify the manuscript according to the review recommendations. Your revisions should address the specific points made by each reviewer.

Yours sincerely,

Mary C Mullins

Academic Editor

PLOS Genetics

Gregory Barsh

Editor-in-Chief

PLOS Genetics

Reviewer's Responses to Questions

**Comments to the Authors:**

Reviewer #1: Authors have revised many parts of the manuscript and presented their novel findings more clearly. There are only a few minor points.

1. Line 530: “…suggesting impaired vitellogenic growth of follicles in the triple knockout with the loss of inhbaa.” Authors may mean “inha” instead of “inhbaa”. The expression level of inhbaa in the triple mutant compared with other mutants or control are not shown in this manuscript.

2. Line 1381:"the number in the box indicates the sample size.", The boxes are absent in Figure 2C. This sentence can be deleted.

3. Figure 9C and D: The data and the statistical analysis are not mentioned in the text. It will be nice to mention the statistical analysis for the data for esr2a, esr2b, fshb and lhb, even when they are not statistically significant.

Reviewer #2: The authors have done a nice job addressing my previous comments.

Reviewer #3: Rereview Zhai Rescue of bmp15 deficiency in zebrafish by mutation of inha

reveals mechanisms of BMP15 regulation of folliculogenesis

In general, the authors’ responses to the critiques are acceptable. They didn’t always do what the reviewers asked, but either responded in some other acceptable way or argued appropriately for another solution.

In making the revision, a number of other generally minor problems arose, as detailed below, which should be fixed.

33 Interestingly, the blockade of folliculogenesis and sex reversal in bmp15 mutantS could

In addition, saying that the blockade is rescued makes it seem that oogenesis is normal in the double mutant. Saying ‘partially rescued’ would correspond better to the results.

53 Bone morphogenetic protein 15 (BMP15) is a protein produced by egg cells in animals

Not all animals.

69 a developing oocyte and surrounding somatic follicle cells (granulosa and theca cells)

Theca cells are not generally considered to be follicle cells.

127 flounder ovarian cell line suppressed expression of steroidogenic genes including

aromatase (cyp19a1a), in contrast to gdf9 [41].

It’s unclear whether gdf9 acts different than bmp15 or cyp19a1a in this sentence construction.

133 Disruption of bmp15 gene in zebrafish resulted in follicle blockade at

previtellogenic (PV) stage followed by sex reversal from females to males.

Give citation.

145 mutantS (bmp15-/-) were arrested at later stage

Interestingly, lrp1ab, lrp5 and lrp6

175 were all expressed in both follicle cells and oocytes whereas lrp2a was exclusively

expressed in the follicle cells (Fig. S1A).

What does that finding say about the likelihood these proteins are actual Vtg receptors?

212 cortical alveoli appear first as a single layer of small vesicles (PV-I; 143±2 μm

The sentence says that the vesicles are 143µm in diameter. I suspect that is incorrect. Say specifically it is the oocyte or the total follicle that you are measuring.

316 To understand how mutation of inha could PARTIALLY rescue the phenotypes of bmp15 mutant, we

Please fix this throughout the text.

317 performed a transcriptome analysis on PG follicles from WT (bmp15+/+;inha+/+),

Was RNA-seq performed on follicles only? Or on complete ovaries? If the former, then if you say ‘dissected follicles’, there’s no ambiguity.

326 genotypes (bmp15-/-, inha-/- and bmp15-/-;inha-/-) and

These two genotypes are identical, so that’s confusing.

326 e similarity between inha-/- and bmp15-/-;inha-/-

It would be better, instead of just saying inha-/-, the text would say bmp15+/+;inha-/-. This would help throughout the text when there’s a mix of double and single mutants being discussed together.

331 marco, nr1d2a and lrp2a, which were down-regulated in bmp15-/- follicles but upregulated

Marco is a macrophage gene. So if the follicles were dissected, these macrophages were apparently infiltrating the follicle. Does that deserve a comment?

434 at PG stage after18 mpf [67].

Also, tell whether there was ovotestis, sex reversal.

474 of the genes examined were significantly up-regulated in the ovary of inha-/- mutant

Since follicles are being investigated, not whole ovaries, it’s probably better to say ‘follicles’ here than ‘ovary’, even though both are probably true.

**Have all data underlying the figures and results presented in the manuscript been provided?**

Reviewer #1: Yes

Reviewer #2: Yes

Reviewer #3: Yes

PLOS authors have the option to publish the peer review history of their article (what does this mean?). If published, this will include your full peer review and any attached files.

Reviewer #1: No

Reviewer #2: No

Reviewer #3: No

---

## [Editor Report · Decision Letter 2]

1 Sep 2023

Dear Dr Ge,

We are pleased to inform you that your manuscript entitled "Rescue of bmp15 deficiency in zebrafish by mutation of inha reveals mechanisms of BMP15 regulation of folliculogenesis" has been editorially accepted for publication in PLOS Genetics. Congratulations!

Yours sincerely,

Mary C Mullins

Academic Editor

PLOS Genetics

Gregory Barsh

Editor-in-Chief

PLOS Genetics

Comments from the reviewers (if applicable):

**Data Deposition**

http://datadryad.org/submit?journalID=pgenetics&manu=PGENETICS-D-23-00186R2

**Press Queries**

---

## [Editor Report · Acceptance letter]

12 Sep 2023

PGENETICS-D-23-00186R2 

 Rescue of  * bmp15 *  deficiency in zebrafish by mutation of  * inha *  reveals mechanisms of BMP15 regulation of folliculogenesis 

Dear Dr Ge, 

We are pleased to inform you that your manuscript entitled " Rescue of  * bmp15 *  deficiency in zebrafish by mutation of  * inha *  reveals mechanisms of BMP15 regulation of folliculogenesis " has been formally accepted for publication in PLOS Genetics! Your manuscript is now with our production department and you will be notified of the publication date in due course.

With kind regards,

Katalin Szabo

PLOS Genetics

On behalf of:
